# Extracting Crop Spatial Distribution from Gaofen 2 Imagery Using a Convolutional Neural Network

Yan Chen [1], Chengming Zhang [1,2,3,*], Shouyi Wang [1], Jianping Li [4], Feng Li [5], Xiaoxia Yang [1,3], Yuanyuan Wang [1,3] and Leikun Yin [1]

[1] College of Information Science and Engineering, Shandong Agricultural University, 61 Daizong Road, Taian 271000, China
[2] Key Open Laboratory of Arid Climate Change and Disaster Reduction of CMA, 2070 Donggangdong Road, Lanzhou 730020, China
[3] Shandong Technology and Engineering Center for Digital Agriculture, 61 Daizong Road, Taian 271000, China
[4] Key Laboratory for Meteorological Disaster Monitoring and Early Warning and Risk Management of Characteristic Agriculture in Arid Regions, CMA, 71 Xinchangxi Road, Yinchuan 750002, China
[5] Shandong Provincal Climate Center, NO.12 Wuying Mountain Road, Jinan 250001, China
*   Correspondence: chming@sdau.edu.cn; Tel.: +86-139-5382-3659

**Abstract:** Using satellite remote sensing has become a mainstream approach for extracting crop spatial distribution. Making edges finer is a challenge, while simultaneously extracting crop spatial distribution information from high-resolution remote sensing images using a convolutional neural network (CNN). Based on the characteristics of the crop area in the Gaofen 2 (GF-2) images, this paper proposes an improved CNN to extract fine crop areas. The CNN comprises a feature extractor and a classifier. The feature extractor employs a spectral feature extraction unit to generate spectral features, and five coding-decoding-pair units to generate five level features. A linear model is used to fuse features of different levels, and the fusion results are up-sampled to obtain a feature map consistent with the structure of the input image. This feature map is used by the classifier to perform pixel-by-pixel classification. In this study, the SegNet and RefineNet models and 21 GF-2 images of Feicheng County, Shandong Province, China, were chosen for comparison experiment. Our approach had an accuracy of 93.26%, which is higher than those of the existing SegNet (78.12%) and RefineNet (86.54%) models. This demonstrates the superiority of the proposed method in extracting crop spatial distribution information from GF-2 remote sensing images.

**Keywords:** convolutional neural network; high-resolution remote sensing imagery; Gaofen 2 imagery; crops; winter wheat; spatial distribution information; Feicheng county

## 1. Introduction

The availability of timely and accurate crop spatial distribution information for large areas is very important for scientific research and agricultural management [1,2]. Traditionally, this information has been obtained through large-scale field surveys. Although this method yields high-precision results, it is time-consuming and labor-intensive [3,4]. Remote sensing technology quickly provides land surface information covering large areas. The pixel-by-pixel classification of remote sensing images is an effective approach to obtain crop spatial distribution information for large areas [5,6]. The technique for extraction of more effective pixel features from remote sensing images is key for improving the accuracy of pixel-based classification [7–9].

Statistically significant spectral and textural features have previously aided in remote sensing image classification problems [10–12]. A remote sensing index is a formulated spectral feature obtained

from mathematical operations on spectral bands, reflecting the information in pixels. The purpose of a remote sensing index is to highlight the information of certain land use types in the image. Frequently employed remote sensing indices include vegetation [13–23], water [24], building [25], and ecology [26]. Better commonly utilized indices include environmental vegetation [13,14], ratio vegetation [15], difference vegetation [16,19], greenness vegetation [17], perpendicular vegetation [18,23], soil adjusted vegetation [20], and normalized difference vegetation [21,22].

The operation rules of the formulated spectral feature are relatively simple and reflect simple statistical information. Consequently, researchers have developed a series of methods for automatically extracting spectral features using computers, including supervised classification methods such as the parallelepiped [27], the minimum distance [27], the Markov distance [28], and the maximum likelihood [29], and non-supervised classification methods, such as the k-means [30].

Textural features describe local patterns and rules on their arrangement in an image, that better describe the spatial correlation of adjacent pixels. Techniques for the extraction of textural information include the Gabor [31], the Gray-Level Co-Occurrence Matrix [32], the Markov random field [33], and the Gibbs random field [34] models, and wavelet transforms [35,36]. The spectral and the textural features are often combined in practical classification because each provides limited image feature information.

The use of high-resolution remote sensing imagery significantly improves the precision and spatial resolution of the extraction results. As traditional feature extraction methods have difficulty extracting effective pixel features from high-resolution remote sensing imagery, researchers have begun using machine learning technology, such as neural networks (NN) [37,38] and support vector machines (SVM) to extract effective pixel features [39,40]. As both NN and SVM are shallow-learning algorithms, they also have difficulty extracting high-level semantic features.

Compared with NN and SVM, the convolutional neural network (CNN) has the ability to extract high-level semantic features. Standard CNNs work in an "image-label" manner, and they output the classes of every image. AlexNet [41], GoogLeNet [42], the Visual Geometry Group (VGG) Network [43], and ResNet [44] are typical standard CNNs. All these models exhibit state-of-the-art image classification performance [45,46]. A "per-pixel-label" model, named the Fully Convolutional Network (FCN), was proposed in 2015 [47]. The FCN utilizes convolutional layers to extract features and employs deconvolutional layers to decode the feature map and restore it to the size of the input image. After FCN was developed, researchers designed a series of convolution-based models to perform pixel-by-pixel classification, such as SegNet [48], DeepLab [49], UNet [50], and RefineNet [51]. SegNet and UNet are clearly structured and are easy to understand. DeepLab employs "Atrous Convolution" to process more detailed images. RefineNet employs a structure termed multi-path refinement to combine coarse high-level features with fine low-level features in order to produce high-resolution semantic segmentation images.

Per-pixel-label models based on CNN have yielded remarkable results in the field of remote sensing image segmentation, and conditional random fields have often been used to further refine the segmentation results [52–55]. Researchers have also established a number of new per-pixel-label models based on CNN, such as the Multi-Scale FCN [56], patch-based CNN [57], Two-Branch CNN [58], CloudNet [59], DCNNGF [60], Multilayer Prediction-CNN [61], patch-wise CNN [62], Cascade CNN [63], and ResFCN [64], to adapt the characteristics of remote sensing images. Convolution-based per-pixel-label models have also enabled the effective extraction of agricultural information from remote sensing images including the extraction of crop plant information [65,66], land-use [67], roads [68], and water [69]. These models have also served in the targeted detection of vehicles [70], ships [56], buildings [71], trees [72,73], weeds [74], and diseases [75].

The remote sensing index method and the SVM method focus on extracting features from the spectral information of pixels themselves, but cannot express the spatial relationship between pixels. Texture features have the ability to express spatial relationships between pixels that are statistically significant. The advantage of CNN is that it can extract the spatial correlation between pixels; however,

the features showing this spatial correlation are obtained through training, and the extraction quality of the features greatly varies by sample. In addition, differences exist between the feature values of the inner pixel and those of the edge pixel for the same object; the edges obtained using CNN are often coarse. The fusion of low-level fine features and high-level rough features to form new features is effective technology for refining these edges.

In crop planting areas in Gaofen 2 (GF-2) images, a pixel covers areas containing dozens of crops, pixel contents are often similar, and image texture is relatively smooth. Based on these characteristics, this paper proposes an improved CNN structure, which we call the crop extraction model (CEM). We used this CEM to extract crop spatial distribution information from GF-2 remote sensing imagery and compared the results with those from the comparison models.

## 2. Study Area and Data

### 2.1. Study Area

Feicheng county is located in central Shandong Province, China (35°53′–36°19′ N; 116°28′–116°59′ E; Figure 1). The plain is the most important terrain of the county, accounting for 44.9% of the total area, followed by mountain, accounting for 33.6%, then hilly area, accounting for 19.9%, lowland regions only accounting for 1.6%. The main food crops are winter wheat and corn. Because the county's geographical and agricultural situation are representative of China, we selected it as our study area to perform the comparison experiments.

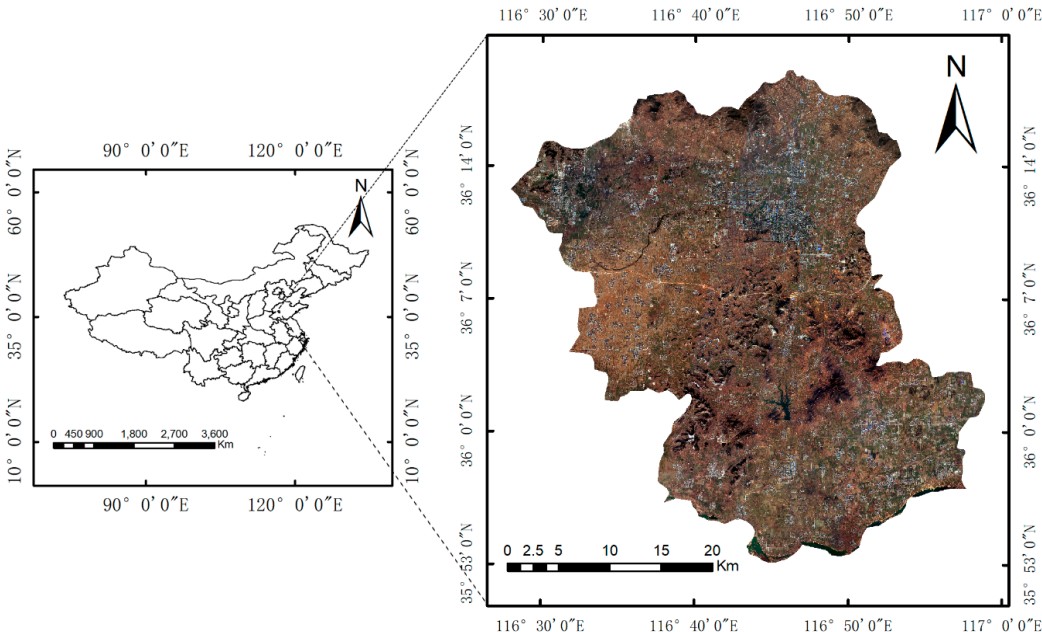

**Figure 1.** The spatial location of Feicheng county and remote sensing images of the study area.

### 2.2. Remote Sensing Imagery

We used 21 GF-2 images covering Feicheng County, including six obtained on 20 February 2017, seven on 8 December 2017, three on 15 February 2018, and five on 16 March 2018. Each GF-2 image comprised a multispectral image and a panchromatic image. Each GF-2 image is divided into a multispectral and panchromatic image. The former is composed of four spectral bands (blue, green, red, and near-infrared), and the spatial resolution of each multispectral image is 4 m, whereas that of the panchromatic image is 1 m.

We designed a program using Python to perform geometric correction. The control points used during this stage were obtained from worldview-4 images that had undergone correction and which had a spatial resolution of 0.3 m. We used the Environment for Visualizing Images (ENVI) software

(Harris Geospatial Solutions, Broomfield, CO, USA) to perform radiometric calibration, atmospheric correction, and image fusion. The parameter used in radiometric calibration stage is published in CRESDA [76]. The Fast Line-of-Sight Atmospheric Analysis of Spectral Hypercubes (FLAASH) model in ENVI was used to conduct atmospheric correction with the Interactive Data Language, which can co-optation better with ENVI. Finally, the pan-sharpening method was used to fuse the multispectral and the panchromatic images. After preprocessing, the spatial resolution of the fused images was 1 m. The fused images were comprised of four bands, namely the blue, green, red, and near-infrared bands.

## 2.3. Establishment of Image-Label Dataset

There are eight main land cover types in the images described in Section 2.2: developed land, bare fields, farmland, agricultural buildings, roads, water bodies, winter wheat, and woodland. Of these, developed land, bare fields, farmland, agricultural buildings, roads, and water bodies can be directly distinguished by visual interpretation in the ENVI software. In order to accurately distinguish between winter wheat and woodland, 391 sample points were located on the images, as shown in Figure 2. Of these, 135 sample points were from woodland areas and 256 were from winter wheat areas. As can be seen in Figure 2, the texture of the woodland area is coarse, the color changes greatly, and the shape is irregular. The texture of the winter wheat area is fine, relatively smooth, and the shape is generally regular. Awareness of these characteristics can help improve the accuracy of visual interpretation.

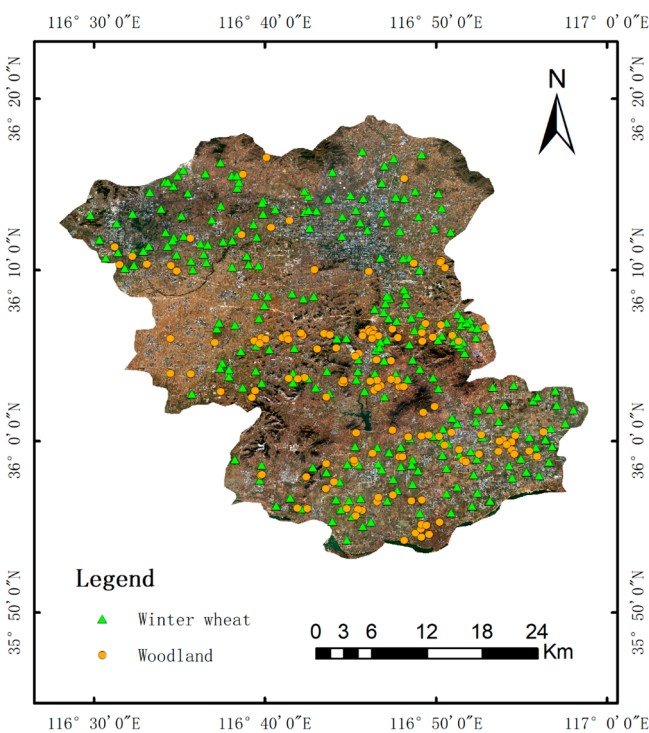

**Figure 2.** The distribution of sampling points acquired via ground investigation.

In total, 537 non-overlapping sub-images were selected from the fused GF-2 remote sensing images described in Section 2.2 to establish the image-label dataset to train the CEM model and test it. The size of each image was 960 × 720 pixels. The dataset covered all land cover types in Feicheng county during the period corresponding to the GF-2 remote sensing images we selected.

We created a label file for each image to record its category number. Each pixel was assigned a code value in the label file. Codes 1–8 were used to represent each land type in the label files. The task of labeling each pixel was performed using the visual interpretation function of ENVI. Figure 3 shows an image and its corresponding label.

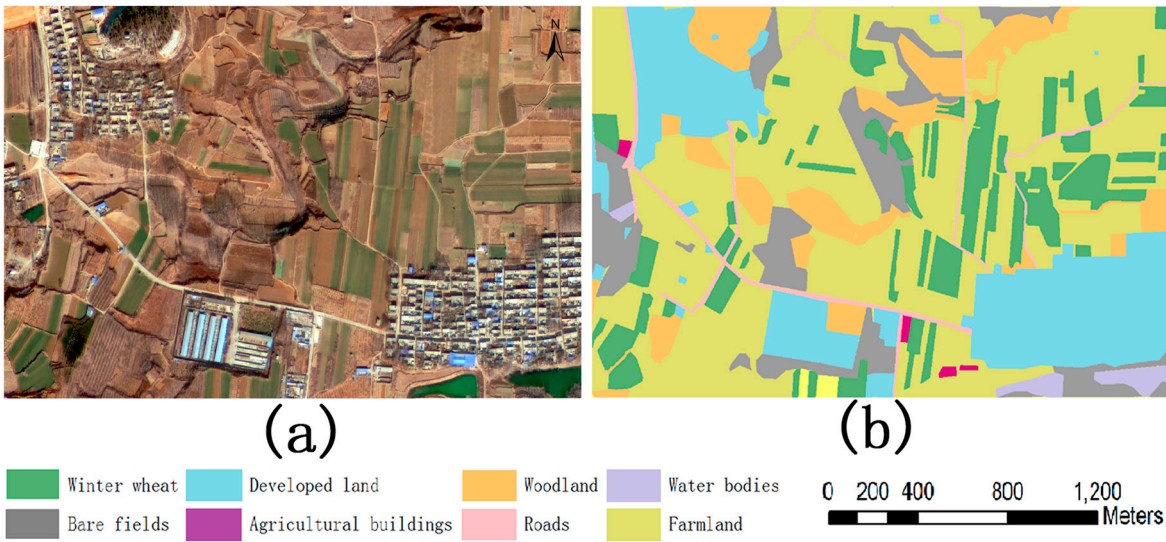

**Figure 3.** Example of image classification: (**a**) original Gaofen 2 image and (**b**) image classified by land use type.

## 3. Methodology

### 3.1. Structure of the Proposed CEM

The CEM model consists of a feature extractor and classifier; the feature extractor comprises an encoder and decoder (Figure 4), the fused GF-2 image and the corresponding label file were used as input. The band order of the image is red, blue, green, and near-infrared.

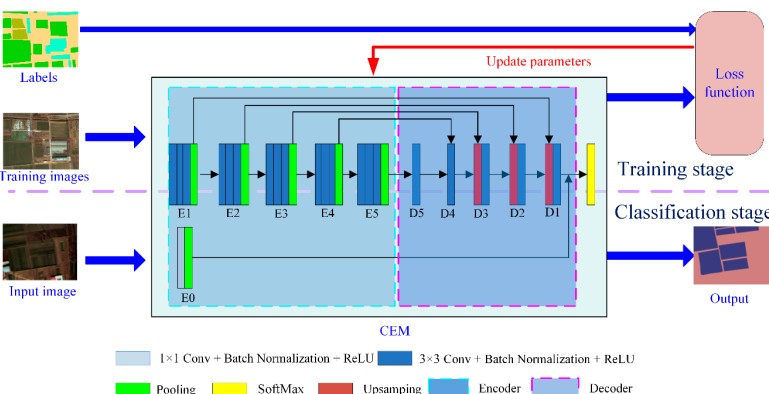

**Figure 4.** Structure of the proposed crop extraction model (CEM) model. ReLU: rectified linear unit; D: Decoder; E: Encoder.

The encoder is composed of a spectral feature extraction unit and five semantic feature extraction units. In Figure 4, E0 represents the spectral feature extraction unit, and E1, E2, E3, E4, and E5 represent semantic feature extraction units, respectively. The word semantic means "similar category information"; in other words, the feature extraction unit extracts similar feature values from pixels belonging to the same category.

The spectral feature extraction unit uses 16 1 × 1-type convolution kernels, including a fixed convolution kernel and 15 ordinary convolution kernels, capable of extracting 16 features from the spectral information of a pixel. The value of the fixed convolution kernel is represented by the vector [0, 0, 0, x], and only the last component is adjusted during the training process. The purpose of the design is to fully exploit the sensitivity of the near-infrared band to vegetation. The image row and

the column structure are unaltered by the spectral feature extraction unit. Therefore, we used this structure as the basic feature for the fusion of each feature level.

Each semantic feature extraction unit includes three feature extraction layers and one pooling layer. Each feature extraction layer has a convolutional layer, a batch normalization layer, and an activation layer for extracting semantic features. The convolutional layers are comprised of $3 \times 3$-type convolution kernels. Table 1 presents the number of convolution kernels for each convolutional layer. The activation layer uses the rectified linear unit function as the activation function.

**Table 1.** Number of convolution kernels for each convolutional layer.

| Layer | Number of Kernels |
|:---:|:---:|
| 1,2,3 | 64 |
| 4,5,6, | 128 |
| 7,8,9 | 256 |
| 10,11,12,13,14,15 | 512 |

The pooling layer was used to optimize the features, but because the ordinary $2 \times 2$ pooling kernel reduces the resolution of the image, we adopted a new pooling strategy. As there are a greater number of pixels for the same land use type in the early feature extraction stages, we used a $2 \times 2$ type pooling kernel in the 1st, 2nd, and 3rd feature extraction units. The use of the $2 \times 2$ type pooling kernel potentially accelerates feature aggregation. In the later stages of feature extraction, the resolution differs significantly from that of the original image, so we adjusted the step size to 1 in the 4th and 5th units. In the 4th and 5th feature extraction units, the size of the pooling kernel is still $2 \times 2$, but when the size of the feature block participating in the pooling operation is smaller than the size of the pooling kernel, the size of the pooling kernel is adjusted to match the feature block; this ensures that valid pooling values are obtained. Therefore, the 4th and 5th feature extraction units do not change the size of the feature map.

The decoder is also composed of five units, as shown by D1, D2, D3, D4, and D5 in Figure 5. The decoder up-samples the feature map to gradually restore the feature image to the size of the original image. The decoder generates feature vectors with the same structure for each pixel.

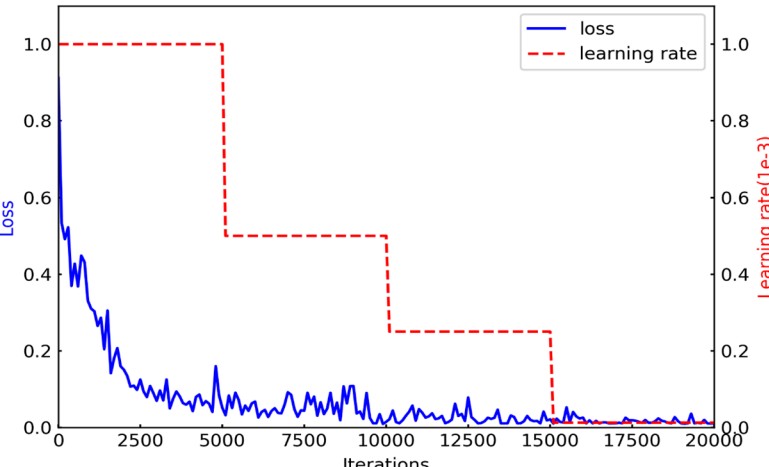

**Figure 5.** Loss rate curve.

The D5 and D4 units are composed of convolution layers. The convolution layer, which works in the 'same' mode in Tensonflow, is used to adjust feature values without changing the size of the feature map. The D5 unit directly uses the output of the E5 unit as input. The input of the D4 unit is the output of the D5 unit and the output of the E4 unit. The inputs are first fused using Equation (1) and then adjusted using the convolution layer.

$$f = a * d5 + b * e4 \tag{1}$$

In Equation (1), f denotes the fused feature map, d5 denotes the output of the D5 unit, and e4 denotes the output of the E4 unit; a and b are fusion coefficients, and the values of a and b are determined after training.

The convolution layer of D4 adjusts the depth of the fused feature map to 256. The D3 unit is composed of an up-sampling layer and a convolutional layer. The up-sampling layer is used to up-sample the output of the D4 unit to match the size of output of the E3 unit; Equation (1) is then used for fusion, but the fusion coefficients are different from those used in D4. The fused features are also adjusted through the convolutional layer. The structures of the D2 and D1 units are the same as that of the D3 unit, and the same working mode is adopted. Table 2 presents the depth of the feature map generated by each decoder unit.

**Table 2.** Depth of feature map generated by each decoder unit.

| Decoder Unit | Depth of Feature Map |
|:---:|:---:|
| D5 | 512 |
| D4 | 512 |
| D3 | 256 |
| D2 | 128 |
| D1 | 64 |

Finally, the semantic features generated by the D1 unit are concatenated with the spectral features outputted by E0 to form a feature map.

The CEM model uses the SoftMax model as a classifier. The SoftMax model exploits the fused feature map generated by the feature extractor to produce the category probability vector for each pixel, and then uses the category corresponding to the maximum value of the probability vector as the category of the pixel.

*3.2. Training the CEM Model*

We employed cross entropy [48–51,77] to define the loss function of the CEM model. Equation (2) illustrates the definition of the cross entropy of a sample:

$$H(p,q) = -\sum_{i=1}^{8} q_i \log(p_i) \tag{2}$$

where $p$ denotes the category probability distribution predicted by the CEM model, $q$ denotes the actual distribution, and $i$ presents the index of an element in a category probability distribution.

On this basis, we defined the loss function of the CEM model as Equation (3):

$$loss = -\frac{1}{ts} \sum_{ts} \sum_{i=1}^{8} q_i \log(p_i) \tag{3}$$

where $ts$ is the number of samples used to train the CEM model.

The CEM model was trained in an end-to-end manner, using six steps, as follows:

1. The parameters of the CEM model were initialized.
2. The selected image-label pairs, which act as the training dataset, were input into the CEM model.
3. Forward propagation of sample images was performed.
4. Loss was calculated and then back-propagation to the CEM model was performed.
5. The stochastic gradient descent was employed to update the network parameters.
6. Re-iteration of steps 3–5 was done until the loss was below the predetermined threshold values.

## 4. Experimental Results and Evaluation

### 4.1. Experimental Setup

The Python 3.6 and the TensorFlow framework were employed on a Linux Ubuntu 16.04 operating system to implement the proposed CEM model. A workstation equipped with a 12 GB NVIDIA Graphics card was used to perform the comparison experiments.

SegNet [48] and RefineNet [51] are state-of-the-art classic pixel-by-pixel semantic image segmentation models for camera images. As the working principles of SegNet and RefineNet are similar to that of the CEM model, we employed these as comparison models to better reflect the superiority of feature extraction and classification of our model.

Since the main crop in Feicheng County during winter is winter wheat, we selected the winter wheat area as the target for experiments.

Total 375 sub-images were selected from the 537 images described in Section 2.3 to compose the training data set, 106 as validation data set, and the remaining 56 as test data set. The training data set, validation set, and test set include all land use types, respectively. Every image in the training data set was processed with color adjustment, horizontal flip, and vertical flip amongst others, to increase the number and diversity of the samples. The color adjustment factors included brightness, saturation, hue, and contrast. After this preprocessing, the final training dataset comprised 4125 images. Table 3 shows the number of samples used in the experiment.

**Table 3.** Number of samples used in the experiment.

| Category | Number of Samples in Training Data Set (Million) | Number of Samples in Validation Data Set (Million) | Number of Samples in Test Data Set (Million) |
|---|---|---|---|
| Winter wheat | 710 | 18 | 9 |
| Agricultural buildings | 2 | 0.1 | 0.07 |
| Woodland | 256 | 6 | 3 |
| Developed land | 542 | 13 | 7 |
| Roads | 23 | 0.6 | 0.3 |
| Water bodies | 25 | 0.7 | 0.3 |
| Farmland | 687 | 17 | 9 |
| Bare fields | 602 | 15 | 8 |

### 4.2. Results of Comparison Experiments

Figure 5 presents the CEM loss rate curve for training. As can be seen from the figure, as the number of iterations increases, the loss value decreases and it eventually tends to stabilize. The loss value decreases rapidly during the first 5000 iterations. From iteration 5000 to 15,000, the loss value fluctuates. After 15,000 iterations, the loss value tends to become stable. The initial learning rate set in the training phase is 0.0001, and the learning rate is reduced to half the existing value every 5000 iterations. It can be seen that as the learning rate decreases, the training loss value also slightly decreases within a certain range. This proves that the learning rate gradient has a beneficial effect on the training network.

In the experiment of this study, it took 37 h to complete the training and 0.02 s to test an image.

Figure 6 presents the extraction results of our model and those of two other models used for comparison with ours. Considering that the goal of this study was to extract crop planting areas, we use Figure 7 to show the results in which the other seven land use types are combined. We use code 9 to present the non-winter wheat. In Figures 6 and 7, a total of 10 sub-images and their extraction results are shown. When selecting these 10 images, we ensured that each image contains winter wheat planting areas and that the other seven land use types appear at least once in an image. This is to enable us to visually compare the algorithms with these results. The first five images are dated 20 February 2017 and the other five are dated 16 March 2018.

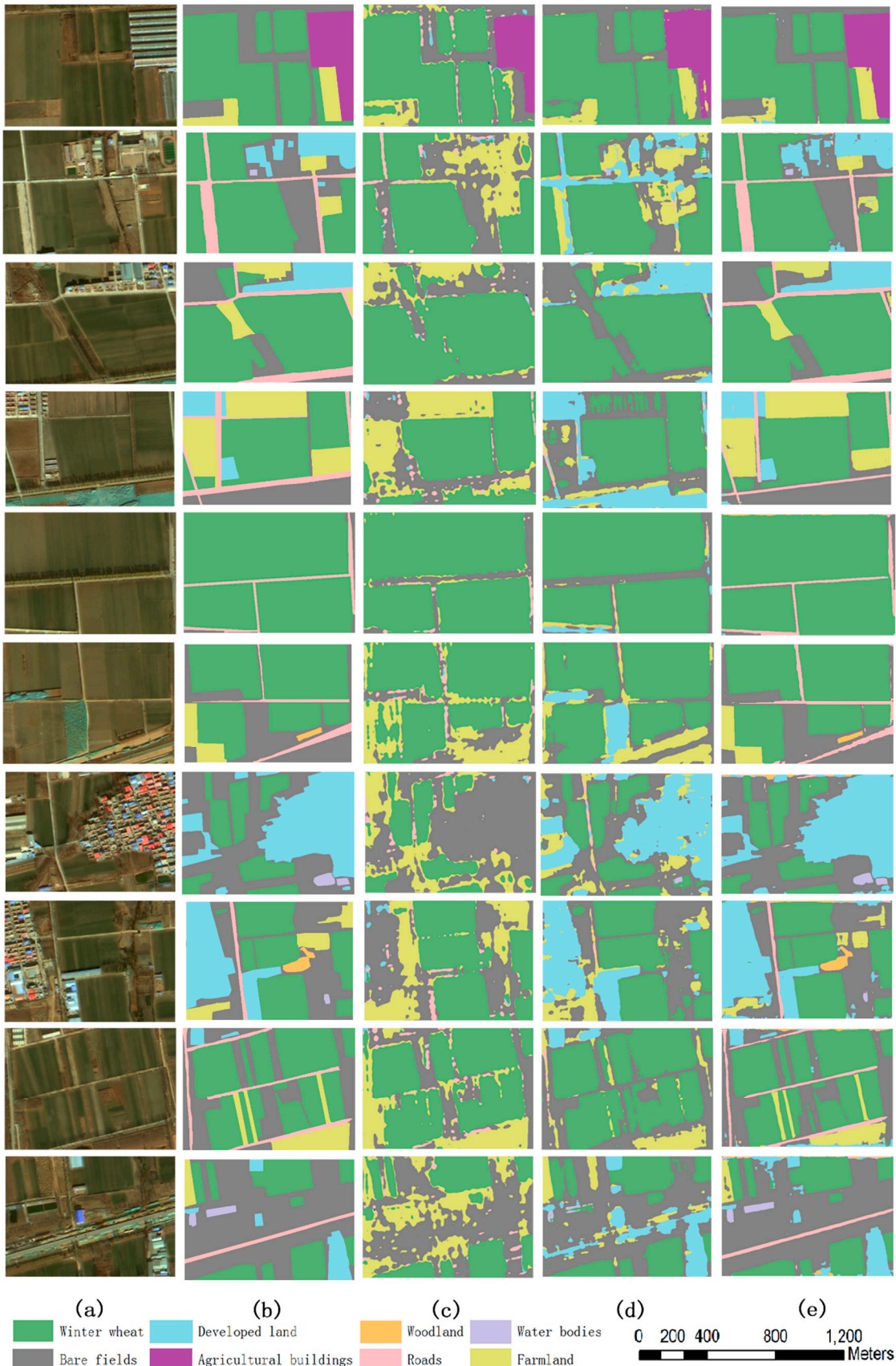

**Figure 6.** Comparison of segmentation results for Gaofen 2 imagery: (**a**) original images, (**b**) ground truth, (**c**) SegNet results, (**d**) RefineNet results, and (**e**) CEM results.

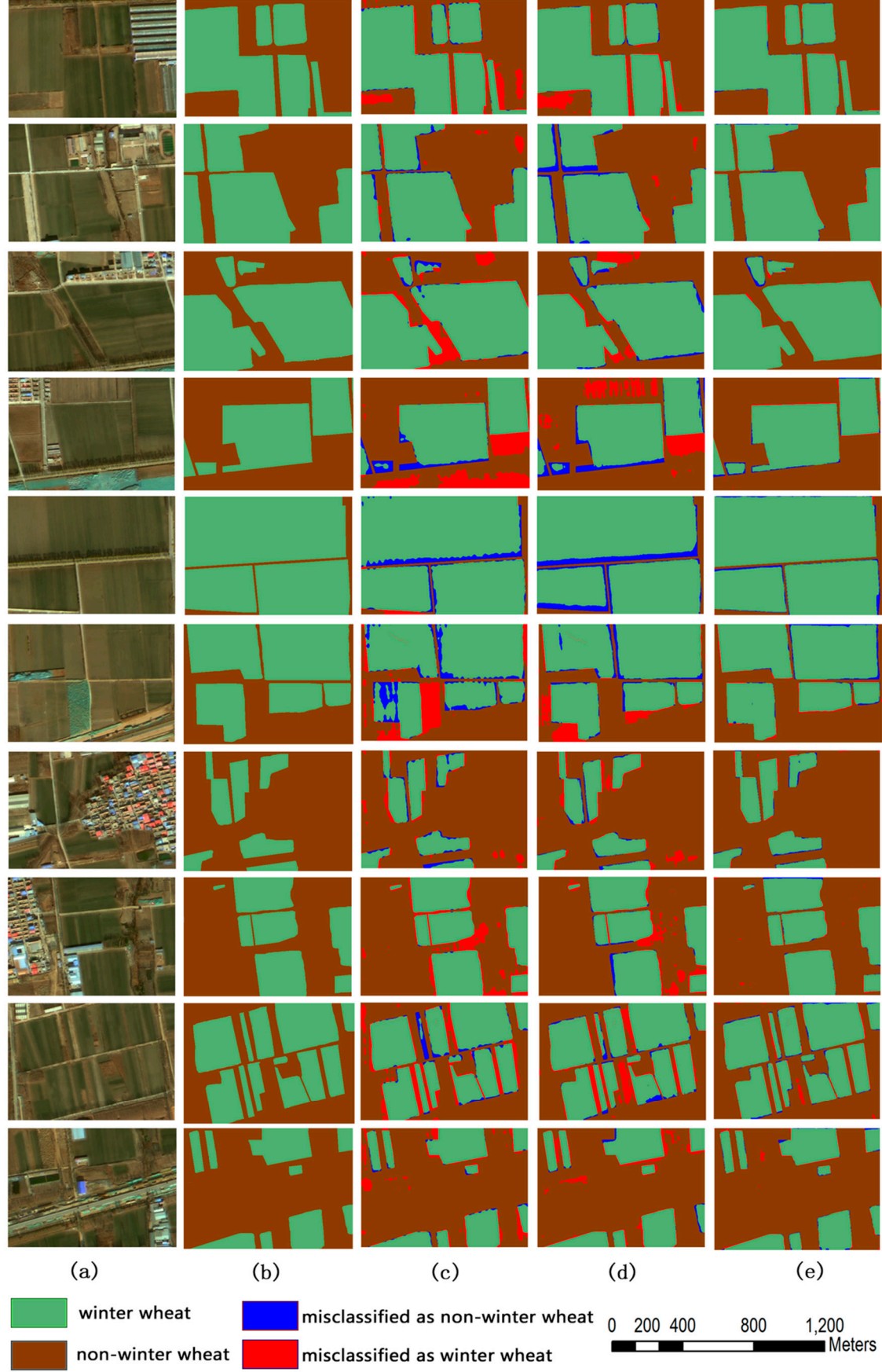

**Figure 7.** Comparison of segmentation results for Gaofen 2 imagery: (**a**) original images, (**b**) ground truth, (**c**) SegNet results, (**d**) RefineNet results, and (**e**) CEM results.

Overall, our algorithm shows better performance with the 10 images than that of the other algorithms. Our algorithm can extract winter wheat planting areas of different scales and shapes, and the extracted crop areas are complete and continuous under complex image environments (Figures 6 and 7). In particular, it can be seen from Figures 6 and 7 that in the region where a larger amount of winter wheat grows, the shape extracted by the proposed algorithm agrees well with the actual region; in contrast, other algorithms have more errors at the edges (red areas in the three columns c, d, e). Because our algorithm uses a combination of semantic and spectral features, there is almost no speckle and noise in the extraction results for the crop planting area, and the edge area extraction improvement over other algorithms is also notable. However, in the edge regions, some errors have still appeared, implying that it is necessary to introduce more information to further improve the extraction results.

Tables 4–6 present confusion matrices for the segmentation results from the three models. Each row of the confusion matrix represents the proportion of the actual category, while each column represents the proportion of the predicted category. As can be seen from Tables 4–6, the proposed CEM model achieved better extraction results. On average, only 1.76% of "winter wheat" pixels were wrongly classified as "non-winter wheat", and 0.89% of "non-winter wheat" pixels were wrongly classified as "winter wheat". Compared with the RefineNet model, this is a reduction of 0.3% and 2.36%, respectively. Compared with the SegNet model, this is a reduction by 2.33% and 2.57%, respectively.

**Table 4.** Confusion matrix for the CEM models.

| Predicted | Winter Wheat | Agricultural Buildings | Woodland | Developed Land | Roads | Water Bodies | Farmland | Bare Fields |
|---|---|---|---|---|---|---|---|---|
| Winter wheat | 23.15% | 0.00% | 0.16% | 0.03% | 0.01% | 0.08% | 0.97% | 0.51% |
| Agricultural buildings | 0.00% | 0.08% | 0.00% | 0.01% | 0.00% | 0.00% | 0.00% | 0.00% |
| Woodland | 0.05% | 0.00% | 8.87% | 0.03% | 0.01% | 0.01% | 0.03% | 0.03% |
| Developed land | 0.02% | 0.01% | 0.00% | 17.37% | 0.02% | 0.00% | 0.53% | 0.41% |
| Roads | 0.02% | 0.00% | 0.00% | 0.04% | 0.72% | 0.00% | 0.02% | 0.03% |
| Water bodies | 0.09% | 0.00% | 0.01% | 0.00% | 0.00% | 0.83% | 0.01% | 0.00% |
| Farmland | 0.38% | 0.00% | 0.15% | 0.38% | 0.02% | 0.00% | 22.18% | 0.90% |
| Bare fields | 0.33% | 0.00% | 0.36% | 0.19% | 0.02% | 0.00% | 0.87% | 20.06% |

**Table 5.** Confusion matrix for the Segnet models.

| Predicted | Winter Wheat | Agricultural Buildings | Woodland | Developed Land | Roads | Water Bodies | Farmland | Bare Fields |
|---|---|---|---|---|---|---|---|---|
| Winter wheat | 20.82% | 0.00% | 1.13% | 1.80% | 0.06% | 0.06% | 0.77% | 0.27% |
| Agricultural buildings | 0.01% | 0.05% | 0.00% | 0.01% | 0.01% | 0.00% | 0.00% | 0.01% |
| Woodland | 0.85% | 0.00% | 6.25% | 0.00% | 0.00% | 0.01% | 0.91% | 1.01% |
| Developed land | 0.73% | 0.04% | 0.00% | 14.47% | 0.11% | 0.00% | 1.15% | 1.86% |
| Roads | 0.03% | 0.00% | 0.00% | 0.01% | 0.69% | 0.00% | 0.09% | 0.01% |
| Water bodies | 0.09% | 0.00% | 0.05% | 0.03% | 0.00% | 0.67% | 0.08% | 0.02% |
| Farmland | 0.56% | 0.00% | 0.01% | 0.01% | 0.12% | 0.00% | 18.34% | 4.97% |
| Bare fields | 1.19% | 0.00% | 0.00% | 0.02% | 0.02% | 0.00% | 3.77% | 16.83% |

**Table 6.** Confusion matrix for the RefineNet models.

| Predicted | Winter Wheat | Agricultural Buildings | Woodland | Developed Land | Roads | Water Bodies | Farmland | Bare Fields |
|---|---|---|---|---|---|---|---|---|
| Winter wheat | 22.95% | 0.00% | 0.68% | 0.01% | 0.06% | 0.08% | 0.84% | 0.29% |
| Agricultural buildings | 0.00% | 0.07% | 0.00% | 0.01% | 0.01% | 0.00% | 0.00% | 0.00% |
| Woodland | 0.43% | 0.00% | 6.59% | 0.05% | 0.00% | 0.00% | 0.07% | 1.89% |
| Developed land | 0.00% | 0.03% | 0.31% | 16.76% | 0.02% | 0.00% | 1.03% | 0.21% |
| Roads | 0.01% | 0.01% | 0.00% | 0.12% | 0.61% | 0.00% | 0.02% | 0.06% |
| Water bodies | 0.21% | 0.00% | 0.04% | 0.00% | 0.00% | 0.65% | 0.00% | 0.04% |
| Farmland | 2.25% | 0.01% | 0.03% | 0.01% | 0.12% | 0.00% | 19.53% | 2.06% |
| Bare fields | 0.35% | 0.00% | 0.03% | 0.39% | 0.01% | 0.00% | 1.67% | 19.38% |

In this study, we employed several criteria, including accuracy, precision, recall, and the Kappa coefficient, to evaluate the performance of the proposed model [52]. Table 7 shows data from the evaluation criteria of the three models, with the accuracy of the CEM model being 15.14% and 6.72% above those of SegNet and RefineNet, respectively.

**Table 7.** Comparison of the performance of the three models.

| Index | CEM | SegNet | RefineNet |
|---|---|---|---|
| Accuracy | 93.26% | 78.12% | 86.54% |
| Precision | 91.75% | 74.38% | 80.87% |
| Recall | 92.01% | 76.64% | 82.02% |
| Kappa | 91.64% | 69.53% | 80.54% |

## 5. Discussion

### 5.1. Tuning of the Parameter Settings of the Proposed CEM

When constructing the structure of the CEM model, we borrowed from the classical RefineNet model and the structure of the VGG model. The multipath structure of the RefineNet model can effectively integrate high-and low-level features and improve the feature extraction ability of the model. We implemented the multipath structure as the basic structure of the CEM model. Considering the characteristics of GF-2 remote sensing images, we set up a feature extraction unit using three downscaling and two feature extraction units that do not change scale. This design could ensure that the CEM model extracts enough features.

When designing the structure of the feature extraction unit, we borrowed the structure of the VGG model and adopted three convolution layer superposition strategies. The advantage of this design is it makes it possible to ensure that feature extraction has sufficient receptive domains and can reduce parameters.

Considering that there are a large number of mixed pixels in the GF-2 images, the use of spectral features is beneficial to improve classification accuracy. Two convolution kernels were set for each class to extract spectral features, and a total of 16 convolution kernels were set. Among them, considering that the infrared band in the GF-2 image is sensitive to vegetation, a fixed convolution kernel was set to map the infrared band's ability to respond to vegetation.

The above strategy effectively improved the generalization ability of the model.

### 5.2. The Advantages of CEM

We first analyzed the characteristics of the crop planting area in the GF-2 images and designed the structure of the CEM model based on those characteristics. By adopting two convolution kernels and utilizing the multi-path refinement structure, the network structure of the CEM model is well adapted to the characteristics of GF-2 images. This explains the higher precision obtained from the CEM model.

The field surveys revealed that in the crop planting area, a GF-2 pixel covers an area containing about 350–450 winter wheat or rice plants and approximately 180 maize plants. Since the content of each pixel area showed no significant difference and the texture is relatively smooth, the multi-layer convolution hardly generated more effective features. It even introduces noise, resulting in poor segmentation effects, as presented in Tables 3 and 4, and Figures 6 and 7.

Considering the similarity in the structures of SegNet, RefineNet, and CEM, we introduced the difference between the maximum value of the category probability vector generated by the classifier and the next maximum value as an indicator to further illustrate the advantages of the CEM model structure. This indicator was termed category confidence. Figure 8 illustrates the category confidence distribution for the results of the experiment. The ratios of pixels at a lower confidence level of the comparison models are greater than those of the CEM model. This demonstrates that the feature value composition of the CEM model is superior to those of the SegNet and RefineNet models.

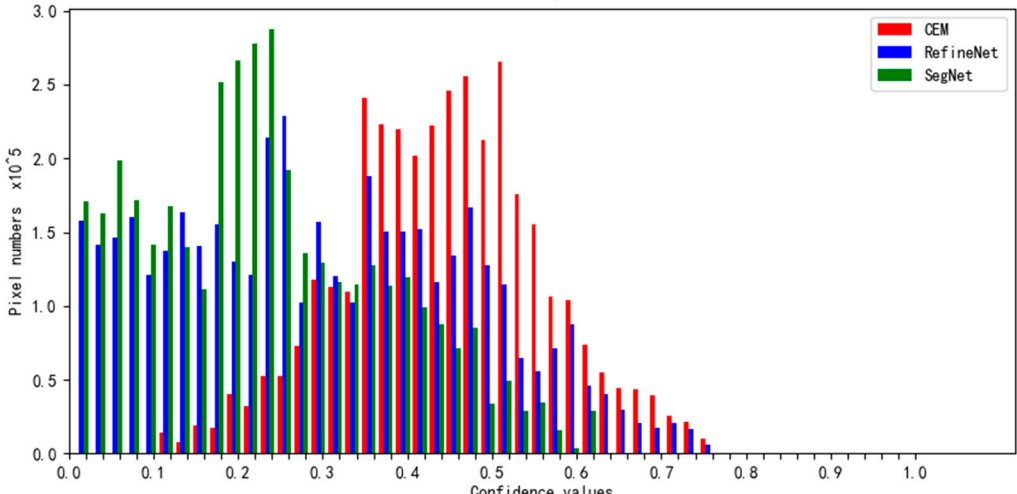

**Figure 8.** The category confidence distribution for the results of the experiment.

### 5.3. SegNet versus CEM

The SegNet model extracts high-level semantic features from rich image details through deep convolution. It demonstrates advantages when extracting objects that cover a large number of pixels. However, if the number of pixels covered by the object is small, or a pixel contains several target objects, this structure not only fails to extract detailed features, but may introduce additional noise due to the expansion of the field of view. This reduces the accuracy of the classification results from the model. When extracting crop spatial distributions from a GF-2 image, although the number of pixels occupied by a farmland is large, the difference between pixels remains small because the area covered by a plant is small, and this obscures the advantages of SegNet.

Contrary to the SegNet model, which extracts 64 high-level semantic features by deepening the hierarchy, the CEM model extracts 80 features by combining spectral features, high-level semantic features, and low-level semantic features. Because CEM fully considers the natural features of crops and the distribution characteristics of farmlands, it is advantageous for identifying pixels at the edges and corners of crop planting areas.

In summary, there are the following differences between CEM and SegNet model.

(1) The SegNet model transforms the highest-level semantic features into pixel feature vectors by step-by-step sampling. Therefore, pixel feature vectors only contain abstract semantic information. The CEM model adopts feature fusion to fuse low-level semantic feature information and high-level semantic information, so the information contained in feature vectors of pixels is more abundant than that of SegNet.

(2) The feature vector generated by the SegNet model has only semantic information. The feature vector generated by the CEM model not only has semantic information, but also statistical information of the spectral values of the pixels themselves.

(3) Although the pooling method adopted by SegNet has the effect of aggregating feature values, but each pooling reduces the size of feature map to 1/4 of the original size, which is not conducive to the generation of pixel feature vectors. The pooling method adopted by CEM not only achieves the purpose of aggregation of feature values, but also reduces the degree of feature map size reduction, which can help to obtain feature vectors with good discrimination.

Figure 9 shows a comparison of the results of the two models. The images in Figure 9 are dated 20 February 2017. As can be seen from Figure 9, only a few pixels are misclassified at the corners in case of the CEM results, while a significantly higher number of pixels are misclassified in the SegNet results, both at the corners and edges, and even within the planting area. In Regions 1 and 2, SegNet's results show more severe jaggedness, and our model has almost no jaggedness. In Regions 3, 4, 5, and 6, SegNet does not make a recognition at all, and our model's recognition effect is very consistent with

the actual regions, indicating that our model overcomes the defects of the SegNet model. Further, the use of only high-level semantic features is an important cause of edge roughness. The use of low-level semantic features and high-level features through fusion is an important way to improve the edge.

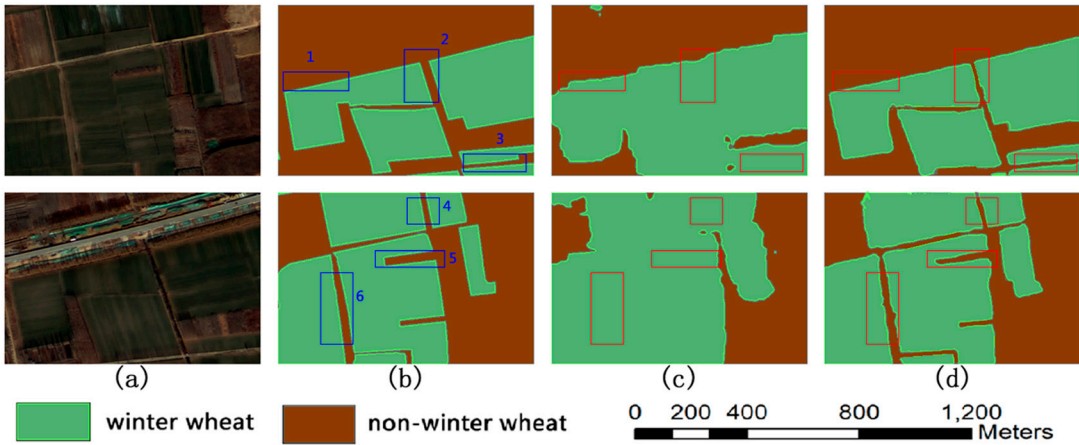

**Figure 9.** Segmentation error comparison: (**a**) original images, (**b**) ground truth, (**c**) errors of SegNet results, and (**d**) errors of CEM results.

### 5.4. RefineNet versus CEM

The RefineNet model also extracts high-level semantic features by deepening the hierarchy. It employs a multi-path refinement structure to fuse low-and high-level semantic features, compensating for missing information in down-sampling. The accuracy of its extraction results is therefore significantly better than that of the SegNet model. However, due to the noise introduced by the deep convolution structure, the edges of the crop planting area in the extraction result remain coarse.

Compared to camera images, remote sensing images reveal fewer details, but with a higher number of channels. Therefore, the structure of the CEM induces advantages including the division of the convolutional layer into five coding units and decoding them separately. The CEM model obtains sufficient information for classification processing while reducing the depth of the convolutional network, by using the convolution kernels of $1 \times 1$ and $3 \times 3$ together. In addition to extracting different level semantic features of a pixel, the CEM model also extracts spectral features. These techniques ensure that the model produces high-precision results.

Figure 10 shows a comparison between the results of the two models. The images in Figure 9 are dated 16 March 2018. As can be seen from Figure 10, only a few pixels are misclassified at the corners of the CEM results, while the RefineNet results show misclassified pixels at the corners and edges of planting areas. Although the number of misclassified pixels in the case of RefineNet is below that for SegNet, it is still significantly higher than that for CEM. In Regions 1, 2, and 4, the RefineNet model classification has many errors, resulting in rough edges, and in the CEM, results show areas in more accurate detail. In Regions 3, 5, and 6, the RefineNet results contain more recognition errors and a considerable amount of winter wheat is not recognized. For the CEM results, the shape obtained is consistent with the actual shapes of areas. Considering the difference in the fusion of high-and low-level semantic features, the strategy adopted in this study is more reasonable than that of RefineNet.

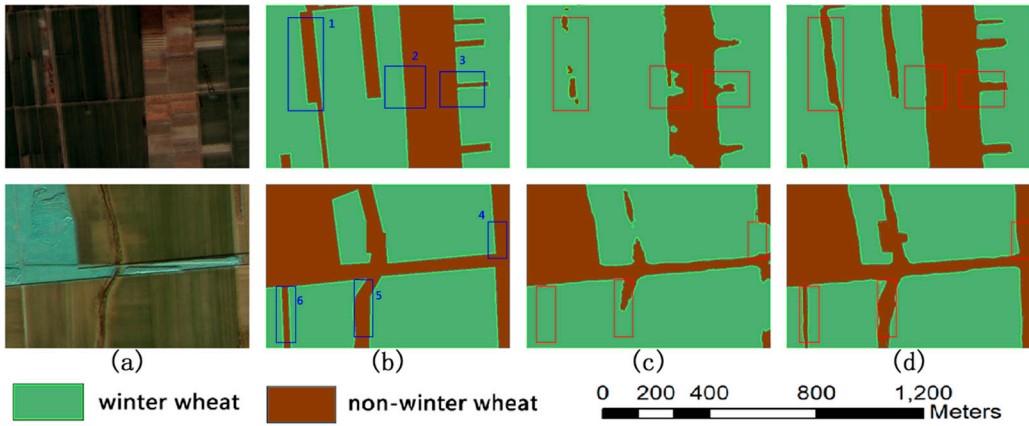

**Figure 10.** Segmentation errors comparison: (**a**) original images, (**b**) ground truth, (**c**) errors of RefineNet results, and (**d**) errors of CEM results.

## 6. Conclusions

Using satellite remote sensing has become a mainstream approach for extracting crop spatial distribution, but field edge results are usually rough, resulting in lowered overall accuracy. In this paper, we proposed a new approach for extracting fine crop spatial distribution information. As can be seen from the comparison experiment, our approach had an accuracy of 93.26%, which is higher than those of the existing SegNet (78.12%) and RefineNet (86.54%) models. The structure of the proposed model can adapt to the characteristics of the crop area in the GF-2 images, which is the key to the success of our model.

The main contributions of this study are as follows: (1) the convolutional network structure designed in this study fully considers the proportional relationship between the area occupied by the target object and that of a pixel in the image. Experiments proved that the proportional relationship and the network structure affect the accuracy of the extraction results, and provide a reference for designing other land use type extraction models. (2) The model uses both $1 \times 1$ type and $3 \times 3$ type convolution kernels to extract pixel features. This reveals the semantic features of the pixels and extracts the spectral features of the pixels. Thus, the advantage of the numerous channels in a remote sensing image fully emerges and compensates for the limitation of fewer details in a remote sensing image.

The main disadvantage of our approach is that it requires more pre-pixel label files. Future research should test the use of semi-supervised classification to reduce the dependence on pre-pixel label files.

**Author Contributions:** Conceptualization: C.Z.; methodology: C.Z. and Y.C.; software: Y.C. and S.W.; validation: J.L., and F.L.; formal analysis: C.Z. and F.L.; investigation: L.Y.; resources: X.Y.; data curation: Y.W.; writing—original draft preparation: C.Z., Y.C. and S.W.; writing—review and editing: C.Z., J.L. and F.L.; visualization: X.Y.; supervision: Y.W.; project administration: C.Z.; funding acquisition: C.Z.

**Funding:** This study was funded by the Science Foundation of Shandong (Grant # ZR2017MD018); the Key Research and Development Program of Ningxia (Grant # 2019BEH03008); the Open Research Project of the Key Laboratory for Meteorological Disaster Monitoring, Early Warning and Risk Management of Characteristic Agriculture in Arid Regions (Grant # CAMF-201701 and CAMF-201803); the arid meteorological science research fund project by the Key Open Laboratory of Arid Climate Change and Disaster Reduction of CMA (Grant # IAM201801).

**Conflicts of Interest:** The authors declare no conflict of interest.

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
