# Peer review of "Extracting Crop Spatial Distribution from Gaofen 2 Imagery Using a Convolutional Neural Network"

_applsci, doi:10.3390/app9142917_

Round 1
Reviewer 1 Report
Dear Authors,
After a very careful study of your work entitled "Extracting Crop Spatial Distribution..." by Yan Chen et al, I have found a really high level work, exhaustive, detailed and complete.The degree of complexity is very high and, in my opinion, the work can be improved in several ways.
Abstract: It can be improved by simplifying the expressions.
Introduction: It is very good but, after a long introduction, one waits a more complete description of the work in the last paragraph
Section 2.3: The paragraph is hard to understand. In addition, Figure 2 must be well explained in its legend and much better called from the text.
Section 2.4: I feel that the actual Figure 4 can be eliminated with no damage to the paper. Its information is quite secondary.
Secion 3.2: The cross-entropy is very important at the time defining the loss function. Give some references please.
Section 4.2 in Figure 6. You have done a lot of work to build it up, however its comments are very few lines. In my opinion, you can extract many more things you think about it. Please, extent the discussion about Figure 6. Dates are also important to compare images on aproximately the same season.
Section 5.2: Include more comments about Figure 8. Dates are equally important than above.
Section 5.4: It must be eliminated. It does not give relevant information
Finally, you describe the computed you use to perform the computations. Please, give some details as the time the program takes to complete the task.
Yours Sincerely
Author Response
Response to reviewer's comments from Applied Sciences (applsci-524108)
Dear Reviewer:
We would like to thank you for the comments and suggestion. We have substantially revised the manuscript and detailed responses are provided below. All revised contents are in blue.
1. Abstract: It can be improved by simplifying the expressions.
Reply: According to your good suggestion, we have revised the abstract. The revised content is as follows:
Using satellite remote sensing has become a mainstream approach for extracting crop spatial distribution. Making edges finer is a challenge while extracting crop spatial distribution information from high-resolution remote sensing images using a convolutional neural network (CNN). Based on the characteristics of the crop area in the Gaofen 2 (GF-2) images, this paper proposes an improved CNN to extract fine crop areas. The CNN comprises a feature extractor and a classifier. The feature extractor employs a spectral feature extraction unit to generate spectral features, and five coding-decoding-pair units to generate five level features. A linear model is used to fuse features of different levels, and the fusion results are up-sampled to obtain a feature map consistent with the structure of the input image. This feature map is used by the classifier to perform pixel-by-pixel classification. In this study, the SegNet and RefineNet models and 21 GF-2 images of Feicheng County, Shandong Province, China, were chosen for comparison experiment. Our approach had an accuracy of 93.26%, which is higher than those of the existing SegNet (78.12%) and RefineNet (86.54%) models. This demonstrates the superiority of the proposed method in extracting crop spatial distribution information from GF-2 remote sensing images.
2. Introduction: It is very good but, after a long introduction, one waits a more complete description of the work in the last paragraph
Reply: According to you good suggestion, we add description content after the introduction, the added content is as follows:
The remote sensing index method and the SVM method focus on extracting features from the spectral information of pixels themselves but cannot express the spatial relationship between pixels. Texture features have the ability to express spatial relationships between pixels that are statistically significant. The advantage of CNN is that it can extract the spatial correlation between pixels; however, the features showing this spatial correlation are obtained through training, and the extraction quality of the features greatly varies by sample. In addition, differences exist between the feature values of the inner pixel and those of the edge pixel for the same object; the edges obtained using CNN are often coarse. The fusion of low-level fine features and high-level rough features to form new features is effective technology for refining these edges.
In crop planting areas in Gaofen 2 (GF-2) images, a pixel covers areas containing dozens of crops, pixel contents are often similar, and image texture is relatively smooth. Based on these characteristics, this paper proposes an improved CNN structure, which we call the crop extraction model (CEM). We used this CEM to extract crop spatial distribution information from GF-2 remote sensing imagery and compared the results with those from the comparison models.
3. Section 2.3: The paragraph is hard to understand. In addition, Figure 2 must be well explained in its legend and much better called from the text.
Reply: According to your good suggestion, we merged Section 2.3 and Section 2.4, the original content of Sec 2.3 had been revised, and we added new content to explain Fig. 2. The revised content is as follows:
2.3 Establishment of Image-label dataset
There are eight main land cover types in the images described in Section 2.2: developed land, bare fields, farmland, agricultural buildings, roads, water bodies, winter wheat, and woodland. Of these, developed land, bare fields, farmland, agricultural buildings, roads, and water bodies can be directly distinguished by visual interpretation in the ENVI software. In order to accurately distinguish between winter wheat and woodland, a total of 391 sample points were located on the images, as shown in Fig. 2. Of these, 135 sample points were from woodland areas and 256 were from winter wheat areas. As can be seen in Fig. 2, the texture of the woodland area is coarse, the color changes greatly, and the shape is irregular. The texture of the winter wheat area is fine, relatively smooth, and the shape is generally regular. Awareness of these characteristics can help improve the accuracy of visual interpretation.
Figure 2. The distribution of sampling points acquired via ground investigation.
In total, 537 non-overlapping sub-images were selected from the fused GF-2 remote sensing images described in Section 2.2 to establish the image-label dataset to train the CEM model and test it. The size of each image was 960×720 pixels. The dataset covered all land cover types in Feicheng county during the period corresponding to the GF-2 remote sensing images we selected.
We created a label file for each image to record its category number. Each pixel was assigned a code value in the label file. Codes 1-8 were used to represent each land type in the label files. The task of labeling each pixel was performed using the visual interpretation function of ENVI. Fig. 3 shows an image and its corresponding label.
Figure 3. Example of image classification: (a) original Gaofen 2 image and (b) image classified by land use type.
4. Section 2.4: I feel that the actual Figure 4 can be eliminated with no damage to the paper. Its information is quite secondary.
Reply: According to your good suggestion, the Fig. 4 had been eliminated.
5. Secion 3.2: The cross-entropy is very important at the time defining the loss function. Give some references please.
Reply: According to your good suggestion, we added the references which proposed the cross-entropy. The revised content is as follows:
We employed cross entropy [48-51, 78] to define the loss function of the CEM model.
78. Bord, P.D.; Kroese, D.P.; Mannor, S.; Rubinstein, R.Y. A Tutorial on the Cross-Entropy Method. Ann. Oper. Res. 2005, 134, doi:19-27, 10.1007/s10479-005-5724-z.
6. Section 4.2 in Figure 6. You have done a lot of work to build it up, however its comments are very few lines. In my opinion, you can extract many more things you think about it. Please, extent the discussion about Figure 6. Dates are also important to compare images on aproximately the same season.
Reply: According to you good suggestion, we revised the relevant content. Firstly, we add content to extent the discussion about Fig. 6. Secondly, we added a paragraph to descript the principle of selecting the images shown in Fig.6. Thirdly, we added the date information of these images in Fig. 6. The revised content is as follows:
4.2 Results of Comparison Experiments
Figure 5 presents the CEM loss rate curve for training. As can be seen from the figure, as the number of iterations increases, the loss value decreases and it eventually tends to stabilize. The loss value decreases rapidly during the first 5000 iterations. From iteration 5000 to 15000, the loss value fluctuates. After 15000 iterations, the loss value tends to become stable. The initial learning rate set in the training phase is 0.0001, and the learning rate is reduced to half the existing value every 5000 iterations. It can be seen that as the learning rate decreases, the training loss value also slightly decreases within a certain range. This proves that the learning rate gradient has benefit effect on the training network.
Figure 5. Loss rate curve.
In the experiment of this study, it took 37 hours to complete the training and 0.02 seconds to test an image.
Figure 6 presents the extraction results of our model and those of two other models used for comparison with ours. Considering that the goal of this study was to extract crop planting areas, we use Figure 7 to show the results in which the other seven land use types are combined. We use code 9 to present the non-winter wheat. In Figures 6 and 7, a total of 10 sub-images and their extraction results are shown. When selecting these 10 images, we ensured that each image contains winter wheat planting areas and that the other seven land use types appears at least once in an image. This is to enable us to visually compare the algorithms with these results. The first five images are dated February 20, 2017 and the other five are dated March 16, 2018.
Overall, our algorithm shows better performance with the 10 images than the other algorithms show. Our algorithm can extract winter wheat planting areas of different scales and shapes, and the extracted crop areas are complete and continuous under complex image environments (Fig. 6, Fig. 7). In particular, it can be seen from Fig .6 and Fig. 7 that in the region where a larger amount of winter wheat grows, the shape extracted by the proposed algorithm agrees well with the actual region; in contrast, other algorithms have more errors at the edges. (red areas in the three columns c, d, e). Because our algorithm uses a combination of semantic and spectral features, there is almost no speckle and noise in the extraction results for the crop planting area, and the edge area extraction improvement over other algorithms is also notable. However, in the edge regions, some errors have still appeared, implying that it is necessary to introduce more information to further improve the extraction results.
Figure 6. Comparison of segmentation results for Gaofen 2 imagery: (a) original images, (b) ground truth, (c) SegNet results, (d) RefineNet results, and (e) CEM results.
Figure 7. Comparison of segmentation results for Gaofen 2 imagery: (a) original images, (b) ground truth, (c) SegNet results, (d) RefineNet results, and (e) CEM results.
Table 4, Table 5 and Table 6 present confusion matrices for the segmentation results from the three models. Each row of the confusion matrix represents the proportion of the actual category, while each column represents the proportion of the predicted category. As can be seen from Table 4, Table 5 and Table 6, the proposed CEM model achieved better extraction results. On average, only 1.76% of “winter wheat” pixels were wrongly classified as “non-winter wheat”, and 0.89% of “non-winter wheat” pixels were wrongly classified as “winter wheat”. Compared with the RefineNet model, this is a reduction of 0.3% and 2.36%, respectively. Compared with the SegNet model, this is a reduction by 2.33% and 2.57%, respectively.
Table 4. Confusion matrix for the CEM models.
Predicted | Winter wheat | Agricultural buildings | Woodland | Developed land | Roads | Water bodies | Farmland | Bare fields |
Winter wheat | 23.15% | 0.00% | 0.16% | 0.03% | 0.01% | 0.08% | 0.97% | 0.51% |
Agricultural buildings | 0.00% | 0.08% | 0.00% | 0.01% | 0.00% | 0.00% | 0.00% | 0.00% |
Woodland | 0.05% | 0.00% | 8.87% | 0.03% | 0.01% | 0.01% | 0.03% | 0.03% |
Developed land | 0.02% | 0.01% | 0.00% | 17.37% | 0.02% | 0.00% | 0.53% | 0.41% |
Roads | 0.02% | 0.00% | 0.00% | 0.04% | 0.72% | 0.00% | 0.02% | 0.03% |
Water bodies | 0.09% | 0.00% | 0.01% | 0.00% | 0.00% | 0.83% | 0.01% | 0.00% |
Farmland | 0.38% | 0.00% | 0.15% | 0.38% | 0.02% | 0.00% | 22.18% | 0.90% |
Bare fields | 0.33% | 0.00% | 0.36% | 0.19% | 0.02% | 0.00% | 0.87% | 20.06% |
Table 5. Confusion matrix for the Segnet models.
Predicted | Winter wheat | Agricultural buildings | Woodland | Developed land | Roads | Water bodies | Farmland | Bare fields |
Winter wheat | 20.82% | 0.00% | 1.13% | 1.80% | 0.06% | 0.06% | 0.77% | 0.27% |
Agricultural buildings | 0.01% | 0.05% | 0.00% | 0.01% | 0.01% | 0.00% | 0.00% | 0.01% |
Woodland | 0.85% | 0.00% | 6.25% | 0.00% | 0.00% | 0.01% | 0.91% | 1.01% |
Developed land | 0.73% | 0.04% | 0.00% | 14.47% | 0.11% | 0.00% | 1.15% | 1.86% |
Roads | 0.03% | 0.00% | 0.00% | 0.01% | 0.69% | 0.00% | 0.09% | 0.01% |
Water bodies | 0.09% | 0.00% | 0.05% | 0.03% | 0.00% | 0.67% | 0.08% | 0.02% |
Farmland | 0.56% | 0.00% | 0.01% | 0.01% | 0.12% | 0.00% | 18.34% | 4.97% |
Bare fields | 1.19% | 0.00% | 0.00% | 0.02% | 0.02% | 0.00% | 3.77% | 16.83% |
Table 6. Confusion matrix for the RefineNet models.
Predicted | Winter wheat | Agricultural buildings | Woodland | Developed land | Roads | Water bodies | Farmland | Bare fields |
Winter wheat | 22.95% | 0.00% | 0.68% | 0.01% | 0.06% | 0.08% | 0.84% | 0.29% |
Agricultural buildings | 0.00% | 0.07% | 0.00% | 0.01% | 0.01% | 0.00% | 0.00% | 0.00% |
Woodland | 0.43% | 0.00% | 6.59% | 0.05% | 0.00% | 0.00% | 0.07% | 1.89% |
Developed land | 0.00% | 0.03% | 0.31% | 16.76% | 0.02% | 0.00% | 1.03% | 0.21% |
Roads | 0.01% | 0.01% | 0.00% | 0.12% | 0.61% | 0.00% | 0.02% | 0.06% |
Water bodies | 0.21% | 0.00% | 0.04% | 0.00% | 0.00% | 0.65% | 0.00% | 0.04% |
Farmland | 2.25% | 0.01% | 0.03% | 0.01% | 0.12% | 0.00% | 19.53% | 2.06% |
Bare fields | 0.35% | 0.00% | 0.03% | 0.39% | 0.01% | 0.00% | 1.67% | 19.38% |
In this study, we employed several criteria, including accuracy, precision, recall, and the Kappa coefficient, to evaluate the performance of the proposed model [52]. Table 7 shows data from the evaluation criteria of the three models, with the accuracy of the CEM model being 15.14% and 6.72% above those of SegNet and RefineNet, respectively.
Table 7. Comparison of the performance of the three models.
Index | CEM | SegNet | RefineNet |
Accuracy | 93.26% | 78.12% | 86.54% |
Precision | 91.75% | 74.38% | 80.87% |
Recall | 92.01% | 76.64% | 82.02% |
Kappa | 91.64% | 69.53% | 80.54% |
7. Section 5.2: Include more comments about Figure 8. Dates are equally important than above.
Reply: According to your good suggestion, we added new comments about Fig.8 and Fig. 9, now named as Fig.9 and Fig. 10, and added the date information of these images in Fig.9 and Fig. 10. The original Section 5.2 now named as 5.3, and original 5.3 as 5.4. The revised content is as follows:
5.3 SegNet versus CEM
The SegNet model extracts high-level semantic features from rich image details through deep convolution. It demonstrates advantages when extracting objects that cover a large number of pixels. However, if the number of pixels covered by the object is small, or a pixel contains several target objects, this structure not only fails to extract detailed features, but may introduce additional noise due to the expansion of the field of view. This reduces the accuracy of the classification results from the model. When extracting crop spatial distributions from a GF-2 image, although the number of pixels occupied by a farmland is large, the difference between pixels remains small because the area covered by a plant is small, and this obscures the advantages of SegNet.
Contrary to the SegNet model, which extracts 64 high-level semantic features by deepening the hierarchy, the CEM model extracts 80 features by combining spectral features, high-level semantic features, and low-level semantic features. Because CEM fully considers the natural features of crops and the distribution characteristics of farmlands, it is advantageous for identifying pixels at the edges and corners of crop planting areas.
Figure 9 shows a comparison of the results of the two models. The images in Fig. 9 are dated February 20, 2017. As can be seen from Figure 9, only a few pixels are misclassified at the corners in case of the CEM results, while a significantly higher number of pixels are misclassified in the SegNet results, both at the corners and edges, and even within the planting area. In Regions 1 and 2, SegNet's results show more severe jaggedness, and our model has almost no jaggedness. In Regions 3, 4, 5, and 6, SegNet does not make a recognition at all, and our model’s recognition effect is very consistent with the actual regions, indicating that our model overcomes the defects of the SegNet model. Further, the use of only high-level semantic features is an important cause of edge roughness. The use of low-level semantic features and high-level features through fusion is an important way to improve the edge.
Figure 9. Segmentation error comparison: (a) original images, (b) ground truth, (c) errors of SegNet results, and (d) errors of CEM results.
5.4 RefineNet versus CEM
The RefineNet model also extracts high-level semantic features by deepening the hierarchy. It employs a multi-path refinement structure to fuse low- and high-level semantic features, compensating for missing information in down-sampling. The accuracy of its extraction results is therefore significantly better than that of the SegNet model. However, due to the noise introduced by the deep convolution structure, the edges of the crop planting area in the extraction result remain coarse.
Compared to camera images, remote sensing images reveal fewer details, but with a higher number of channels. Therefore, the structure of the CEM induces advantages including the division of the convolutional layer into five coding units and decoding them separately. The CEM model obtains sufficient information for classification processing while reducing the depth of the convolutional network, by using convolution kernels of 1×1 and 3×3 together. In addition to extracting different level semantic features of a pixel, the CEM model also extracts spectral features. These techniques ensure that the model produces high-precision results.
Figure 10 shows a comparison between the results of the two models. The images in Fig. 9 are dated March 16, 2018. As can be seen from Figure 10, only a few pixels are misclassified at the corners of the CEM results, while the RefineNet results show misclassified pixels at the corners and edges of planting areas. Although the number of misclassified pixels in the case of RefineNet is below that for SegNet, it is still significantly higher than that for CEM. In Regions 1, 2, and 4, the RefineNet model classification has many errors, resulting in rough edges, and in the CEM results show areas in more accurate detail. In Regions 3, 5, and 6, the RefineNet results contain more recognition errors and a considerable amount of winter wheat is not recognized. For the CEM results, the shape obtained is consistent with the actual shapes of areas. Considering the difference in the fusion of high-and low-level semantic features, the strategy adopted in this study is more reasonable than that of RefineNet.
Figure 10. Segmentation errors comparison: (a) original images, (b) ground truth, (c) errors of RefineNet results, and (d) errors of CEM results.
8.Section 5.4: It must be eliminated. It does not give relevant information
Reply: According to your good suggestion, the original Sec 5.4 had been eliminated.
9. Finally, you describe the computed you use to perform the computations. Please, give some details as the time the program takes to complete the task.
Reply: According to your good suggestion, we add the detail content of the experiment in Section 4.2. The revised content is as follows: 4.2 Results of Comparison Experiments
Figure 5 presents the CEM loss rate curve for training. As can be seen from the figure, as the number of iterations increases, the loss value decreases and it eventually tends to stabilize. The loss value decreases rapidly during the first 5000 iterations. From iteration 5000 to 15000, the loss value fluctuates. After 15000 iterations, the loss value tends to become stable. The initial learning rate set in the training phase is 0.0001, and the learning rate is reduced to half the existing value every 5000 iterations. It can be seen that as the learning rate decreases, the training loss value also slightly decreases within a certain range. This proves that the learning rate gradient has benefit effect on the training network.
Figure 5. Loss rate curve.
In the experiment of this study, it took 37 hours to complete the training and 0.02 seconds to test an image.

Reviewer 2 Report
This paper presents a Convolutional Neural Network method to extract fine crop spatial distribution using Gaofen 2 images. Few comments:
Line 125-126: images with 0.3 m resolution, are they aerial images or satellite images? Please specify.
Line 133: fused images 1m resolution, what is the original resolution? Is it the same? Please clarify.
Line 149: you mentioned 21 GF-2images in Line 121, here 537 images were selected, so are they samples? If so what is the size of original images?
Support the conclusions by some results (give numbers)
Author Response
Response to reviewer's comments from Applied Sciences (applsci-524108)
Dear Reviewer:
We would like to thank you for the comments and suggestion. We have substantially revised the manuscript and detailed responses are provided below. All revised contents are in blue.
1. Line 125-126: images with 0.3 m resolution, are they aerial images or satellite images? Please specify.
Reply: According to your good suggestion, we revised the relevant content, and specify the information of the images. The revised content is as follows:
We designed a program using Python to perform geometric correction. The control points used during this stage were obtained from worldview-4 images that had undergone correction and which had a spatial resolution of 0.3 m.
2. Line 133: fused images 1m resolution, what is the original resolution? Is it the same? Please clarify.
Reply: According to your good suggestion, we revised the relevant content, added description of the original images. The revised content is as follows:
We used 21 GF-2 images covering Feicheng County, including six obtained on February 20, 2017, seven on December 8, 2017, three on February 15, 2018, and five on March 16, 2018. Each GF-2 image comprised a multispectral image and a panchromatic image. Each GF-2 image is divided into a multispectral and panchromatic image. The former is composed of four spectral bands (blue, green, red, and near-infrared), and the spatial resolution of each multispectral image is 4 m, whereas that of the panchromatic image is 1 m.
We designed a program using Python to perform geometric correction. The control points used during this stage were obtained from worldview-4 images that had undergone correction and which had a spatial resolution of 0.3 m. We used the Environment for Visualizing Images (ENVI) software (Harris Geospatial Solutions, Broomfield, Colorado, U.S.A.) to perform radiometric calibration, atmospheric correction, and image fusion. The parameter used in radiometric calibration stage is published in CRESDA [77]. The Fast Line-of-Sight Atmospheric Analysis of Spectral Hypercubes (FLAASH) model in ENVI was used to conduct atmospheric correction with the Interactive Data Language, which can co-optation better with ENVI. Finally, the pan-sharpening method was used to fuse the multispectral and the panchromatic images. After preprocessing, the spatial resolution of the fused images was 1 m. The fused images comprised four bands, namely the blue, green, red, and near-infrared bands.
3. Line 149: you mentioned 21 GF-2images in Line 121, here 537 images were selected, so are they samples? If so what is the size of original images?
Reply: According to your good suggestion, we revised the relevant content. The revised content is as follows:
4.1 Experimental Setup
The Python 3.6 and the TensorFlow framework were employed on a Linux Ubuntu 16.04 operating system to implement the proposed CEM model. A workstation equipped with a 12 GB NVIDIA Graphics card was used to perform the comparison experiments.
SegNet [48] and RefineNet [51] are state-of-the-art classic pixel-by-pixel semantic image segmentation models for camera images. As the working principles of SegNet and RefineNet are similar to that of the CEM model, we employed these as comparison models to better reflect the superiority of feature extraction and classification of our model.
Since the main crop in Feicheng County during winter is winter wheat, we selected the winter wheat area as the target for experiments.
Total 375 sub-images were selected from the 537 images described in Section 2.3 to compose the training data set, 106 as validation data set, and the remaining 56 as test data set. The training data set, validation set, and test set include all land use types, respectively. Every image in the training data set was processed with color adjustment, horizontal flip, and vertical flip amongst others, to increase the number and diversity of the samples. The color adjustment factors included brightness, saturation, hue, and contrast. After this preprocessing, the final training dataset comprised 4,125 images. Table 3 shows the number of samples used in the experiment.
Table 3. Number of samples used in the experiment.
Category | Number of samples in training data set(million) | Number of samples in validation data set(million) | Number of samples in test data set (million) |
Winter wheat | 710 | 18 | 9 |
Agricultural buildings | 2 | 0.1 | 0.07 |
Woodland | 256 | 6 | 3 |
Developed land | 542 | 13 | 7 |
Roads | 23 | 0.6 | 0.3 |
Water bodies | 25 | 0.7 | 0.3 |
Farmland | 687 | 17 | 9 |
Bare fields | 602 | 15 | 8 |
4. Support the conclusions by some results (give numbers)
Reply: According to your good suggestion, we revised the conclusions. The revised content is as follows:
6. Conclusion
Using satellite remote sensing has become a mainstream approach for extracting crop spatial distribution, but field edge results are usually rough, resulting in lowered overall accuracy. In this paper, we proposed a new approach for extracting fine crop spatial distribution information. As can be seen from the comparison experiment, our approach had an accuracy of 93.26%, which is higher than those of the existing SegNet (78.12%) and RefineNet (86.54%) models. The structure of the proposed model can adapt to the characteristics of the crop area in the GF-2 images, which is the key to the success of our model.
The main contributions of this study are as follows: (1) The convolutional network structure designed in this study fully considers the proportional relationship between the area occupied by the target object and that of a pixel in the image. Experiments proved that the proportional relationship and the network structure affect the accuracy of the extraction results, and provide a reference for designing other land use type extraction models. (2) The model uses both 1×1 type and 3×3 type convolution kernels to extract pixel features. This reveals the semantic features of the pixels and extracts the spectral features of the pixels. Thus, the advantage of the numerous channels in a remote sensing image fully emerges and compensates for the limitation of fewer details in a remote sensing image.
The main disadvantage of our approach is that it requires more pre-pixel label files. Future research should test the use of semi-supervised classification to reduce the dependence on pre-pixel label files.

Reviewer 3 Report
The principal objective of the paper is an application of convolutional neural network (CNN) to extract crop spatial distributions using Gaofen 2 (GF-2) images. An experiment of two class classification (winter and non-winter wheat) is included. The possible contribution of the paper is only experimental since CNN is a well-known method that has been applied extensively. The explanations of the proposed method have room for improvement. In addition, the method is proposed for a 9-class classification problem, but it is applied to a 2-class classification problem. Thus, results are not convincing, more experiments should be included. Some and analysis of the solved detection problem could be implemented. Literal presentation of the paper is poor. In summary, I consider the contents, as they are, cannot be published. Specifically, this paper has the following major issues:
- Literal presentation has room for improvement. For instance, (i) Please explain the point of using the term “semantic” in this application. (ii) Check the use of verbal tenses, e.g., page 3, lines 103-109, use present tense for the present work, e.g., “We introduce”; “and establish”. (iii) Page 4, line 149, please explain the term “fused remote sensing images”. (iv) Page 5, lines 163-164, “Each image was processed using 10 …. 5,907 images” change to “After this preprocessing, the final dataset comprised 5,907 images” please avoid verbose. (v) Page 5, lines 174-175, “The CEM model uses the fused high score 2 image and the corresponding marker as input.” What “fused high score 2 image” is? Please be consistent with the use of terms “corresponding marker” labels?. (vi) Figure 5, “Images” or “Training images”? (vii) Page 6, line 194, “aggregates features to optimize the features” ? (viii) Change title of Section 5 to “Discussion”.
- The proposed method is designed for a problem with a number of classes (8-9), but it is applied to a much simpler problem (2 classes). It is absolutely inconsistent, besides the number of arbitrary parameters employed for the proposed method. The results for the two-class problem could be valid, but there are several issues to be solved (see the comments below). However, receiver operating characteristic (ROC) curve analyses are requested and results for the proposed multi-class problem are needed to evaluate the real contribution of the paper.
- There is not clear the number of classes of the proposed problem, 8? 9? Page 4, lines 154-156: “Codes 1-8 were used to represent each land type in the label files for the training stage. In the label file produced by the CEM, code 1 denotes winter wheat, and code 9 denotes non-winter wheat.” Please explain better the number of classes of the problem.
- Page 6 and following pages, explanations of the tuning of the parameter setting is not comprehensive. There an amount of arbitrary fixed values without explanation. A rationale about this should be included. Why five extraction units of semantic feature, 15 ordinary kernels, three feature extraction layers, 16 features…? Have those parameters been tuned to a problem of 9 classes? Explain the generalization capability of the proposed method to other problems.
- Several technical aspects should be addressed: (i) the value of the cost function through iterations should be represented and explained. (ii) Table 2, values of confusion matrices must add to 100%. (iii) The “semantic” features should be represented and discussed. (iv) The discussion of Section 5 should include more supporting numerical data, for instance, page 10, lines 297-299, “more high-level semantic features”, “”more features”,…
Author Response
Response to reviewer's comments from Applied Sciences (applsci-524108)
Dear Reviewer:
We would like to thank you for the comments and suggestion. We have substantially revised the manuscript and detailed responses are provided below. All revised contents are in blue.
1. The principal objective of the paper is an application of convolutional neural network (CNN) to extract crop spatial distributions using Gaofen 2 (GF-2) images. An experiment of two class classification (winter and non-winter wheat) is included. The possible contribution of the paper is only experimental since CNN is a well-known method that has been applied extensively. The explanations of the proposed method have room for improvement. In addition, the method is proposed for a 9-class classification problem, but it is applied to a 2-class classification problem. Thus, results are not convincing, more experiments should be included. Some and analysis of the solved detection problem could be implemented. Literal presentation of the paper is poor. In summary, I consider the contents, as they are, cannot be published. Specifically, this paper has the following major issues:
Reply: According to your good suggestion, we revised the relevant content. Firstly, we revised Section 3.1 to make the explanations of the proposed CEM clear. Secondly, we redesigned and conducted the experiment, based on the new result, we revised the whole Section 4, and Section5. Thirdly, in order to improve the quality of expression, we employ professional English editors to carry out language editing of manuscripts.
The revised Section 3.1 is as follows:
3.1 Structure of the proposed CEM
The CEM model consists of a feature extractor and classifier; the feature extractor comprises an encoder and decoder (Figure 4), The fused GF-2 image and the corresponding label file were used as input. The band order of the image is red, blue, green, and near-infrared.
Figure 4. Structure of the proposed CEM model. ReLU: rectified linear unit.
The encoder composed of a spectral feature extraction unit and five semantic feature extraction units. In Fig. 4, E0 represents the spectral feature extraction unit, and E1, E2, E3, E4, and E5 represent semantic feature extraction units, respectively. The word semantic means “similar category information”; in other words, the feature extraction unit extracts similar feature values from pixels belonging to the same category.
The spectral feature extraction unit uses 16 1×1-type convolution kernels, including a fixed convolution kernel and 15 ordinary convolution kernels, capable of extracting 16 features from the spectral information of a pixel. The value of the fixed convolution kernel is represented by the vector [0, 0, 0, x], and only the last component is adjusted during the training process. The purpose of the design is to fully exploit the sensitivity of the near-infrared band to vegetation. The image row and the column structure are unaltered by the spectral feature extraction unit. Therefore, we used this structure as the basic feature for the fusion of each feature level.
Each semantic feature extraction unit includes three feature extraction layers and one pooling layer. Each feature extraction layer has a convolutional layer, a batch normalization layer, and an activation layer for extracting semantic features. The convolutional layers comprise 3×3-type convolution kernels. Table 1 presents the number of convolution kernels for each convolutional layer. The activation layer uses the rectified linear unit function as the activation function.
Table 1. Number of convolution kernels for each convolutional layer.
Layer | Number of kernels |
1,2,3 | 64 |
4,5,6, | 128 |
7,8,9 | 256 |
10,11,12,13,14,15 | 512 |
The pooling layer was used to optimize the features, but because the ordinary 2×2 pooling kernel reduces the resolution of the image, we adopted a new pooling strategy. As there are a greater number of pixels for the same land use type in the early feature extraction stages, we used a 2×2 type pooling kernel in the 1st, 2nd, and 3rd feature extraction units. The use of the 2×2 type pooling kernel potentially accelerates feature aggregation. In the later stages of feature extraction, the resolution differs significantly from that of the original image, so we adjusted the step size to 1 in the 4th and 5th units. In the 4th and 5th feature extraction units, the size of the pooling kernel is still 2×2, but when the size of the feature block participating in the pooling operation is smaller than the size of the pooling kernel, the size of the pooling kernel is adjusted to match the feature block; this ensures that valid pooling values are obtained. Therefore, the 4th and 5th feature extraction units do not change the size of the feature map.
The decoder is also composed of five units, as shown by D1, D2, D3, D4, and D5 in Fig. 5. The decoder up-samples the feature map to gradually restore the feature image to the size of the original image. The decoder generates feature vectors with the same structure for each pixel.
The D5 and D4 units are composed of convolution layers. The convolution layer, which works in the ‘same’ mode in Tensonflow, is used to adjust feature values without changing the size of the feature map. The D5 unit directly uses the output of the E5 unit as input. The input of the D4 unit is the output of the D5 unit and the output of the E4 unit. The inputs are first fused using Equation 1 and then adjusted using the convolution layer.
(1)
In Equation 1, f denotes the fused feature map, d5 denotes the output of the D5 unit, and e4 denotes the output of the E4 unit. A and b are fusion coefficients, and the values of a and b are determined after training.
The convolution layer of D4 adjust the depth of the fused feature map to 256. The D3 unit is composed of an up-sampling layer and a convolutional layer. The up-sampling layer is used to up sample the output of the D4 unit to match the size of output of the E3 unit; Equation 1 is then used for fusion, but use fusion coefficients is different from those used in D4. The fused features are also adjusted through the convolutional layer. The structures of the D2 and D1 units are the same as that of the D3 unit, and the same working mode is adopted. Table 2 presents the depth of feature map generated by each decoder unit.
Table 2. Depth of feature map generated by each decoder unit.
Decoder unit | Depth of feature map |
D5 | 512 |
D4, | 512 |
D3 | 256 |
D2 | 128 |
D1 | 64 |
Finally, the semantic features generated by the D1 unit are concatenated with the spectral features outputted by E0 to form a feature map.
The CEM model uses the SoftMax model as a classifier. The SoftMax model exploits the fused feature map generated by the feature extractor to produce the category probability vector for each pixel, and then uses the category corresponding to the maximum value of the probability vector as the category of the pixel.
The revised Section 4 is as follows:
4. Experimental Results and Evaluation
4.1 Experimental Setup
The Python 3.6 and the TensorFlow framework were employed on a Linux Ubuntu 16.04 operating system to implement the proposed CEM model. A workstation equipped with a 12 GB NVIDIA Graphics card was used to perform the comparison experiments.
SegNet [48] and RefineNet [51] are state-of-the-art classic pixel-by-pixel semantic image segmentation models for camera images. As the working principles of SegNet and RefineNet are similar to that of the CEM model, we employed these as comparison models to better reflect the superiority of feature extraction and classification of our model.
Since the main crop in Feicheng County during winter is winter wheat, we selected the winter wheat area as the target for experiments.
Total 375 sub-images were selected from the 537 images described in Section 2.3 to compose the training data set, 106 as validation data set, and the remaining 56 as test data set. The training data set, validation set, and test set include all land use types, respectively. Every image in the training data set was processed with color adjustment, horizontal flip, and vertical flip amongst others, to increase the number and diversity of the samples. The color adjustment factors included brightness, saturation, hue, and contrast. After this preprocessing, the final training dataset comprised 4,125 images. Table 3 shows the number of samples used in the experiment.
Table 3. Number of samples used in the experiment.
Category | Number of samples in training data set(million) | Number of samples in validation data set(million) | Number of samples in test data set (million) |
Winter wheat | 710 | 18 | 9 |
Agricultural buildings | 2 | 0.1 | 0.07 |
Woodland | 256 | 6 | 3 |
Developed land | 542 | 13 | 7 |
Roads | 23 | 0.6 | 0.3 |
Water bodies | 25 | 0.7 | 0.3 |
Farmland | 687 | 17 | 9 |
Bare fields | 602 | 15 | 8 |
4.2 Results of Comparison Experiments
Figure 5 presents the CEM loss rate curve for training. As can be seen from the figure, as the number of iterations increases, the loss value decreases and it eventually tends to stabilize. The loss value decreases rapidly during the first 5000 iterations. From iteration 5000 to 15000, the loss value fluctuates. After 15000 iterations, the loss value tends to become stable. The initial learning rate set in the training phase is 0.0001, and the learning rate is reduced to half the existing value every 5000 iterations. It can be seen that as the learning rate decreases, the training loss value also slightly decreases within a certain range. This proves that the learning rate gradient has benefit effect on the training network.
Figure 5. Loss rate curve.
In the experiment of this study, it took 37 hours to complete the training and 0.02 seconds to test an image.
Figure 6 presents the extraction results of our model and those of two other models used for comparison with ours. Considering that the goal of this study was to extract crop planting areas, we use Figure 7 to show the results in which the other seven land use types are combined. We use code 9 to present the non-winter wheat. In Figures 6 and 7, a total of 10 sub-images and their extraction results are shown. When selecting these 10 images, we ensured that each image contains winter wheat planting areas and that the other seven land use types appears at least once in an image. This is to enable us to visually compare the algorithms with these results. The first five images are dated February 20, 2017 and the other five are dated March 16, 2018.
Overall, our algorithm shows better performance with the 10 images than the other algorithms show. Our algorithm can extract winter wheat planting areas of different scales and shapes, and the extracted crop areas are complete and continuous under complex image environments (Fig. 6, Fig. 7). In particular, it can be seen from Fig .6 and Fig. 7 that in the region where a larger amount of winter wheat grows, the shape extracted by the proposed algorithm agrees well with the actual region; in contrast, other algorithms have more errors at the edges. (red areas in the three columns c, d, e). Because our algorithm uses a combination of semantic and spectral features, there is almost no speckle and noise in the extraction results for the crop planting area, and the edge area extraction improvement over other algorithms is also notable. However, in the edge regions, some errors have still appeared, implying that it is necessary to introduce more information to further improve the extraction results.
Figure 6. Comparison of segmentation results for Gaofen 2 imagery: (a) original images, (b) ground truth, (c) SegNet results, (d) RefineNet results, and (e) CEM results.
Figure 7. Comparison of segmentation results for Gaofen 2 imagery: (a) original images, (b) ground truth, (c) SegNet results, (d) RefineNet results, and (e) CEM results.
Table 4, Table 5 and Table 6 present confusion matrices for the segmentation results from the three models. Each row of the confusion matrix represents the proportion of the actual category, while each column represents the proportion of the predicted category. As can be seen from Table 4, Table 5 and Table 6, the proposed CEM model achieved better extraction results. On average, only 1.76% of “winter wheat” pixels were wrongly classified as “non-winter wheat”, and 0.89% of “non-winter wheat” pixels were wrongly classified as “winter wheat”. Compared with the RefineNet model, this is a reduction of 0.3% and 2.36%, respectively. Compared with the SegNet model, this is a reduction by 2.33% and 2.57%, respectively.
Table 4. Confusion matrix for the CEM models.
Predicted | Winter wheat | Agricultural buildings | Woodland | Developed land | Roads | Water bodies | Farmland | Bare fields |
Winter wheat | 23.15% | 0.00% | 0.16% | 0.03% | 0.01% | 0.08% | 0.97% | 0.51% |
Agricultural buildings | 0.00% | 0.08% | 0.00% | 0.01% | 0.00% | 0.00% | 0.00% | 0.00% |
Woodland | 0.05% | 0.00% | 8.87% | 0.03% | 0.01% | 0.01% | 0.03% | 0.03% |
Developed land | 0.02% | 0.01% | 0.00% | 17.37% | 0.02% | 0.00% | 0.53% | 0.41% |
Roads | 0.02% | 0.00% | 0.00% | 0.04% | 0.72% | 0.00% | 0.02% | 0.03% |
Water bodies | 0.09% | 0.00% | 0.01% | 0.00% | 0.00% | 0.83% | 0.01% | 0.00% |
Farmland | 0.38% | 0.00% | 0.15% | 0.38% | 0.02% | 0.00% | 22.18% | 0.90% |
Bare fields | 0.33% | 0.00% | 0.36% | 0.19% | 0.02% | 0.00% | 0.87% | 20.06% |
Table 5. Confusion matrix for the Segnet models.
Predicted | Winter wheat | Agricultural buildings | Woodland | Developed land | Roads | Water bodies | Farmland | Bare fields |
Winter wheat | 20.82% | 0.00% | 1.13% | 1.80% | 0.06% | 0.06% | 0.77% | 0.27% |
Agricultural buildings | 0.01% | 0.05% | 0.00% | 0.01% | 0.01% | 0.00% | 0.00% | 0.01% |
Woodland | 0.85% | 0.00% | 6.25% | 0.00% | 0.00% | 0.01% | 0.91% | 1.01% |
Developed land | 0.73% | 0.04% | 0.00% | 14.47% | 0.11% | 0.00% | 1.15% | 1.86% |
Roads | 0.03% | 0.00% | 0.00% | 0.01% | 0.69% | 0.00% | 0.09% | 0.01% |
Water bodies | 0.09% | 0.00% | 0.05% | 0.03% | 0.00% | 0.67% | 0.08% | 0.02% |
Farmland | 0.56% | 0.00% | 0.01% | 0.01% | 0.12% | 0.00% | 18.34% | 4.97% |
Bare fields | 1.19% | 0.00% | 0.00% | 0.02% | 0.02% | 0.00% | 3.77% | 16.83% |
Table 6. Confusion matrix for the RefineNet models.
Predicted | Winter wheat | Agricultural buildings | Woodland | Developed land | Roads | Water bodies | Farmland | Bare fields |
Winter wheat | 22.95% | 0.00% | 0.68% | 0.01% | 0.06% | 0.08% | 0.84% | 0.29% |
Agricultural buildings | 0.00% | 0.07% | 0.00% | 0.01% | 0.01% | 0.00% | 0.00% | 0.00% |
Woodland | 0.43% | 0.00% | 6.59% | 0.05% | 0.00% | 0.00% | 0.07% | 1.89% |
Developed land | 0.00% | 0.03% | 0.31% | 16.76% | 0.02% | 0.00% | 1.03% | 0.21% |
Roads | 0.01% | 0.01% | 0.00% | 0.12% | 0.61% | 0.00% | 0.02% | 0.06% |
Water bodies | 0.21% | 0.00% | 0.04% | 0.00% | 0.00% | 0.65% | 0.00% | 0.04% |
Farmland | 2.25% | 0.01% | 0.03% | 0.01% | 0.12% | 0.00% | 19.53% | 2.06% |
Bare fields | 0.35% | 0.00% | 0.03% | 0.39% | 0.01% | 0.00% | 1.67% | 19.38% |
In this study, we employed several criteria, including accuracy, precision, recall, and the Kappa coefficient, to evaluate the performance of the proposed model [52]. Table 7 shows data from the evaluation criteria of the three models, with the accuracy of the CEM model being 15.14% and 6.72% above those of SegNet and RefineNet, respectively.
Table 7. Comparison of the performance of the three models.
Index | CEM | SegNet | RefineNet |
Accuracy | 93.26% | 78.12% | 86.54% |
Precision | 91.75% | 74.38% | 80.87% |
Recall | 92.01% | 76.64% | 82.02% |
Kappa | 91.64% | 69.53% | 80.54% |
The revised Section 5 is as follows:
5. Discussion
5.1 Tuning of the parameter settings of the proposed CEM
When constructing the structure of the CEM model, we borrowed from the classical RefineNet model and the structure of the VGG model. The multipath structure of the RefineNet model can effectively integrate high- and low-level features and improve the feature extraction ability of the model. We implemented the multipath structure as the basic structure of the CEM model. Considering the characteristics of GF-2 remote sensing images, we set up a feature extraction unit using three downscaling and two feature extraction units that do not change scale. This design could ensure that the CEM model extracts enough features.
When designing the structure of the feature extraction unit, we borrowed the structure of the VGG model and adopted three convolution layer superposition strategies. The advantage of this design is it makes it possible to ensure that feature extraction has enough receptive domains and can reduce parameters.
Considering that there are a large number of mixed pixels in the GF-2 images, the use of spectral features is beneficial to improve classification accuracy. Two convolution kernels were set for each class to extract spectral features, and a total of 16 convolution kernels were set. Among them, considering that the infrared band in the GF-2 image is sensitive to vegetation, a fixed convolution kernel was set to map the infrared band's ability to respond to vegetation.
The above strategy effectively improved the generalization ability of the model.
5.2 The Advantages of CEM
We first analyzed the characteristics of the crop planting area in the GF-2 images and designed the structure of the CEM model based on those characteristics. By adopting two convolution kernels and utilizing the multi-path refinement structure, the network structure of the CEM model is well adapted to the characteristics of GF-2 images. This explains the higher precision obtained from the CEM model.
The field surveys revealed that in the crop planting area, a GF-2 pixel covers an area containing about 350 - 450 winter wheat or rice plants and about 180 maize plants. Since the content of each pixel area showed no significant difference and the texture is relatively smooth, the multi-layer convolution hardly generated more effective features. It even introduces noise, resulting in poor segmentation effects as presented in Tables 3 and 4, and Figure 6 and 7.
Considering the similarity in the structures of SegNet, RefineNet, and CEM, we introduced the difference between the maximum value of the category probability vector generated by the classifier and the next maximum value as an indicator to further illustrate the advantages of the CEM model structure. This indicator was termed category confidence. Figure 8 illustrates the category confidence distribution for the results of the experiment. The ratios of pixels at a lower confidence level of the comparison models are greater than those of the CEM model. This demonstrates that the feature value composition of the CEM model is superior to those of the SegNet and RefineNet models.
Figure 8. The category confidence distribution for the results of the experiment.
5.3 SegNet versus CEM
The SegNet model extracts high-level semantic features from rich image details through deep convolution. It demonstrates advantages when extracting objects that cover a large number of pixels. However, if the number of pixels covered by the object is small, or a pixel contains several target objects, this structure not only fails to extract detailed features, but may introduce additional noise due to the expansion of the field of view. This reduces the accuracy of the classification results from the model. When extracting crop spatial distributions from a GF-2 image, although the number of pixels occupied by a farmland is large, the difference between pixels remains small because the area covered by a plant is small, and this obscures the advantages of SegNet.
Contrary to the SegNet model, which extracts 64 high-level semantic features by deepening the hierarchy, the CEM model extracts 80 features by combining spectral features, high-level semantic features, and low-level semantic features. Because CEM fully considers the natural features of crops and the distribution characteristics of farmlands, it is advantageous for identifying pixels at the edges and corners of crop planting areas.
Figure 9 shows a comparison of the results of the two models. The images in Fig. 9 are dated February 20, 2017. As can be seen from Figure 9, only a few pixels are misclassified at the corners in case of the CEM results, while a significantly higher number of pixels are misclassified in the SegNet results, both at the corners and edges, and even within the planting area. In Regions 1 and 2, SegNet's results show more severe jaggedness, and our model has almost no jaggedness. In Regions 3, 4, 5, and 6, SegNet does not make a recognition at all, and our model’s recognition effect is very consistent with the actual regions, indicating that our model overcomes the defects of the SegNet model. Further, the use of only high-level semantic features is an important cause of edge roughness. The use of low-level semantic features and high-level features through fusion is an important way to improve the edge.
Figure 9. Segmentation error comparison: (a) original images, (b) ground truth, (c) errors of SegNet results, and (d) errors of CEM results.
5.4 RefineNet versus CEM
The RefineNet model also extracts high-level semantic features by deepening the hierarchy. It employs a multi-path refinement structure to fuse low- and high-level semantic features, compensating for missing information in down-sampling. The accuracy of its extraction results is therefore significantly better than that of the SegNet model. However, due to the noise introduced by the deep convolution structure, the edges of the crop planting area in the extraction result remain coarse.
Compared to camera images, remote sensing images reveal fewer details, but with a higher number of channels. Therefore, the structure of the CEM induces advantages including the division of the convolutional layer into five coding units and decoding them separately. The CEM model obtains sufficient information for classification processing while reducing the depth of the convolutional network, by using convolution kernels of 1×1 and 3×3 together. In addition to extracting different level semantic features of a pixel, the CEM model also extracts spectral features. These techniques ensure that the model produces high-precision results.
Figure 10 shows a comparison between the results of the two models. The images in Fig. 9 are dated March 16, 2018. As can be seen from Figure 10, only a few pixels are misclassified at the corners of the CEM results, while the RefineNet results show misclassified pixels at the corners and edges of planting areas. Although the number of misclassified pixels in the case of RefineNet is below that for SegNet, it is still significantly higher than that for CEM. In Regions 1, 2, and 4, the RefineNet model classification has many errors, resulting in rough edges, and in the CEM results show areas in more accurate detail. In Regions 3, 5, and 6, the RefineNet results contain more recognition errors and a considerable amount of winter wheat is not recognized. For the CEM results, the shape obtained is consistent with the actual shapes of areas. Considering the difference in the fusion of high-and low-level semantic features, the strategy adopted in this study is more reasonable than that of RefineNet.
Figure 10. Segmentation errors comparison: (a) original images, (b) ground truth, (c) errors of RefineNet results, and (d) errors of CEM results.
2. Literal presentation has room for improvement. For instance, (i) Please explain the point of using the term “semantic” in this application. (ii) Check the use of verbal tenses, e.g., page 3, lines 103-109, use present tense for the present work, e.g., “We introduce”; “and establish”. (iii) Page 4, line 149, please explain the term “fused remote sensing images”. (iv) Page 5, lines 163-164, “Each image was processed using 10 …. 5,907 images” change to “After this preprocessing, the final dataset comprised 5,907 images” please avoid verbose. (v) Page 5, lines 174-175, “The CEM model uses the fused high score 2 image and the corresponding marker as input.” What “fused high score 2 image” is? Please be consistent with the use of terms “corresponding marker” labels?. (vi) Figure 5, “Images” or “Training images”? (vii) Page 6, line 194, “aggregates features to optimize the features” ? (viii) Change title of Section 5 to “Discussion”.
Reply: According to your good suggestion, we employ professional English editors to carry out language editing to improve the quality of expression. Specially, the point-by-point response to what you have highlighted is as follows.
(i) Please explain the point of using the term “semantic” in this application.
We added the explanation content of term “semantic” in the Section 3.1. The added content is as follows:
The encoder composed of a spectral feature extraction unit and five semantic feature extraction units. In Fig. 4, E0 represents the spectral feature extraction unit, and E1, E2, E3, E4, and E5 represent semantic feature extraction units, respectively. The word semantic means “similar category information”; in other words, the feature extraction unit extracts similar feature values from pixels belonging to the same category.
(ii) Check the use of verbal tenses, e.g., page 3, lines 103-109, use present tense for the present work, e.g., “We introduce”; “and establish”.
We have corrected the wrong verbal tenses. The revised content is as follows:
In crop planting areas in Gaofen 2 (GF-2) images, a pixel covers areas containing dozens of crops, pixel contents are often similar, and image texture is relatively smooth. Based on these characteristics, this paper proposes an improved CNN structure, which we call the crop extraction model (CEM). We used this CEM to extract crop spatial distribution information from GF-2 remote sensing imagery and compared the results with those from the comparison models.
(iii) Page 4, line 149, please explain the term “fused remote sensing images”.
We have revised the relevant content to make the expression clear. The revised content is as follows:
In total, 537 non-overlapping sub-images were selected from the fused GF-2 remote sensing images described in Section 2.2 to establish the image-label dataset to train the CEM model and test it. The size of each image was 960×720 pixels. The dataset covered all land cover types in Feicheng county during the period corresponding to the GF-2 remote sensing images we selected.
(iv) Page 5, lines 163-164, “Each image was processed using 10 …. 5,907 images” change to “After this preprocessing, the final dataset comprised 5,907 images” please avoid verbose.
We have revised the relevant content according to your good suggestion.
(v) Page 5, lines 174-175, “The CEM model uses the fused high score 2 image and the corresponding marker as input.” What “fused high score 2 image” is? Please be consistent with the use of terms “corresponding marker” labels?.
We have revised the relevant content to correct these mistake. The revised content is as follows:
The CEM model consists of a feature extractor and classifier; the feature extractor comprises an encoder and decoder (Figure 4), The fused GF-2 image and the corresponding label file were used as input. The band order of the image is red, blue, green, and near-infrared.
(vi) Figure 5, “Images” or “Training images”?
We have revised the Fig. 5. The revised content is as follows:
(vii) Page 6, line 194, “aggregates features to optimize the features” ?
We have revised the relevant content to make the expression clear. The revised content is as follows:
The pooling layer was used to optimize the features, but because the ordinary 2×2 pooling kernel reduces the resolution of the image, we adopted a new pooling strategy. As there are a greater number of pixels for the same land use type in the early feature extraction stages, we used a 2×2 type pooling kernel in the 1st, 2nd, and 3rd feature extraction units. The use of the 2×2 type pooling kernel potentially accelerates feature aggregation. In the later stages of feature extraction, the resolution differs significantly from that of the original image, so we adjusted the step size to 1 in the 4th and 5th units. In the 4th and 5th feature extraction units, the size of the pooling kernel is still 2×2, but when the size of the feature block participating in the pooling operation is smaller than the size of the pooling kernel, the size of the pooling kernel is adjusted to match the feature block; this ensures that valid pooling values are obtained. Therefore, the 4th and 5th feature extraction units do not change the size of the feature map.
(viii) Change title of Section 5 to “Discussion”.
We have change title of Section 5 to “Disscussion” according to you good suggestion.
3. The proposed method is designed for a problem with a number of classes (8-9), but it is applied to a much simpler problem (2 classes). It is absolutely inconsistent, besides the number of arbitrary parameters employed for the proposed method. The results for the two-class problem could be valid, but there are several issues to be solved (see the comments below). However, receiver operating characteristic (ROC) curve analyses are requested and results for the proposed multi-class problem are needed to evaluate the real contribution of the paper.
Reply: According to your good suggestion, we redesigned and conducted the experiment, based on the new result, we revised the whole Section 4. The detail of revised Section4 is as follows:
4. Experimental Results and Evaluation
4.1 Experimental Setup
The Python 3.6 and the TensorFlow framework were employed on a Linux Ubuntu 16.04 operating system to implement the proposed CEM model. A workstation equipped with a 12 GB NVIDIA Graphics card was used to perform the comparison experiments.
SegNet [48] and RefineNet [51] are state-of-the-art classic pixel-by-pixel semantic image segmentation models for camera images. As the working principles of SegNet and RefineNet are similar to that of the CEM model, we employed these as comparison models to better reflect the superiority of feature extraction and classification of our model.
Since the main crop in Feicheng County during winter is winter wheat, we selected the winter wheat area as the target for experiments.
Total 375 sub-images were selected from the 537 images described in Section 2.3 to compose the training data set, 106 as validation data set, and the remaining 56 as test data set. The training data set, validation set, and test set include all land use types, respectively. Every image in the training data set was processed with color adjustment, horizontal flip, and vertical flip amongst others, to increase the number and diversity of the samples. The color adjustment factors included brightness, saturation, hue, and contrast. After this preprocessing, the final training dataset comprised 4,125 images. Table 3 shows the number of samples used in the experiment.
Table 3. Number of samples used in the experiment.
Category | Number of samples in training data set(million) | Number of samples in validation data set(million) | Number of samples in test data set (million) |
Winter wheat | 710 | 18 | 9 |
Agricultural buildings | 2 | 0.1 | 0.07 |
Woodland | 256 | 6 | 3 |
Developed land | 542 | 13 | 7 |
Roads | 23 | 0.6 | 0.3 |
Water bodies | 25 | 0.7 | 0.3 |
Farmland | 687 | 17 | 9 |
Bare fields | 602 | 15 | 8 |
4.2 Results of Comparison Experiments
Figure 5 presents the CEM loss rate curve for training. As can be seen from the figure, as the number of iterations increases, the loss value decreases and it eventually tends to stabilize. The loss value decreases rapidly during the first 5000 iterations. From iteration 5000 to 15000, the loss value fluctuates. After 15000 iterations, the loss value tends to become stable. The initial learning rate set in the training phase is 0.0001, and the learning rate is reduced to half the existing value every 5000 iterations. It can be seen that as the learning rate decreases, the training loss value also slightly decreases within a certain range. This proves that the learning rate gradient has benefit effect on the training network.
Figure 5. Loss rate curve.
In the experiment of this study, it took 37 hours to complete the training and 0.02 seconds to test an image.
Figure 6 presents the extraction results of our model and those of two other models used for comparison with ours. Considering that the goal of this study was to extract crop planting areas, we use Figure 7 to show the results in which the other seven land use types are combined. We use code 9 to present the non-winter wheat. In Figures 6 and 7, a total of 10 sub-images and their extraction results are shown. When selecting these 10 images, we ensured that each image contains winter wheat planting areas and that the other seven land use types appears at least once in an image. This is to enable us to visually compare the algorithms with these results. The first five images are dated February 20, 2017 and the other five are dated March 16, 2018.
Overall, our algorithm shows better performance with the 10 images than the other algorithms show. Our algorithm can extract winter wheat planting areas of different scales and shapes, and the extracted crop areas are complete and continuous under complex image environments (Fig. 6, Fig. 7). In particular, it can be seen from Fig .6 and Fig. 7 that in the region where a larger amount of winter wheat grows, the shape extracted by the proposed algorithm agrees well with the actual region; in contrast, other algorithms have more errors at the edges. (red areas in the three columns c, d, e). Because our algorithm uses a combination of semantic and spectral features, there is almost no speckle and noise in the extraction results for the crop planting area, and the edge area extraction improvement over other algorithms is also notable. However, in the edge regions, some errors have still appeared, implying that it is necessary to introduce more information to further improve the extraction results.
Figure 6. Comparison of segmentation results for Gaofen 2 imagery: (a) original images, (b) ground truth, (c) SegNet results, (d) RefineNet results, and (e) CEM results.
Figure 7. Comparison of segmentation results for Gaofen 2 imagery: (a) original images, (b) ground truth, (c) SegNet results, (d) RefineNet results, and (e) CEM results.
Table 4, Table 5 and Table 6 present confusion matrices for the segmentation results from the three models. Each row of the confusion matrix represents the proportion of the actual category, while each column represents the proportion of the predicted category. As can be seen from Table 4, Table 5 and Table 6, the proposed CEM model achieved better extraction results. On average, only 1.76% of “winter wheat” pixels were wrongly classified as “non-winter wheat”, and 0.89% of “non-winter wheat” pixels were wrongly classified as “winter wheat”. Compared with the RefineNet model, this is a reduction of 0.3% and 2.36%, respectively. Compared with the SegNet model, this is a reduction by 2.33% and 2.57%, respectively.
Table 4. Confusion matrix for the CEM models.
Predicted | Winter wheat | Agricultural buildings | Woodland | Developed land | Roads | Water bodies | Farmland | Bare fields |
Winter wheat | 23.15% | 0.00% | 0.16% | 0.03% | 0.01% | 0.08% | 0.97% | 0.51% |
Agricultural buildings | 0.00% | 0.08% | 0.00% | 0.01% | 0.00% | 0.00% | 0.00% | 0.00% |
Woodland | 0.05% | 0.00% | 8.87% | 0.03% | 0.01% | 0.01% | 0.03% | 0.03% |
Developed land | 0.02% | 0.01% | 0.00% | 17.37% | 0.02% | 0.00% | 0.53% | 0.41% |
Roads | 0.02% | 0.00% | 0.00% | 0.04% | 0.72% | 0.00% | 0.02% | 0.03% |
Water bodies | 0.09% | 0.00% | 0.01% | 0.00% | 0.00% | 0.83% | 0.01% | 0.00% |
Farmland | 0.38% | 0.00% | 0.15% | 0.38% | 0.02% | 0.00% | 22.18% | 0.90% |
Bare fields | 0.33% | 0.00% | 0.36% | 0.19% | 0.02% | 0.00% | 0.87% | 20.06% |
Table 5. Confusion matrix for the Segnet models.
Predicted | Winter wheat | Agricultural buildings | Woodland | Developed land | Roads | Water bodies | Farmland | Bare fields |
Winter wheat | 20.82% | 0.00% | 1.13% | 1.80% | 0.06% | 0.06% | 0.77% | 0.27% |
Agricultural buildings | 0.01% | 0.05% | 0.00% | 0.01% | 0.01% | 0.00% | 0.00% | 0.01% |
Woodland | 0.85% | 0.00% | 6.25% | 0.00% | 0.00% | 0.01% | 0.91% | 1.01% |
Developed land | 0.73% | 0.04% | 0.00% | 14.47% | 0.11% | 0.00% | 1.15% | 1.86% |
Roads | 0.03% | 0.00% | 0.00% | 0.01% | 0.69% | 0.00% | 0.09% | 0.01% |
Water bodies | 0.09% | 0.00% | 0.05% | 0.03% | 0.00% | 0.67% | 0.08% | 0.02% |
Farmland | 0.56% | 0.00% | 0.01% | 0.01% | 0.12% | 0.00% | 18.34% | 4.97% |
Bare fields | 1.19% | 0.00% | 0.00% | 0.02% | 0.02% | 0.00% | 3.77% | 16.83% |
Table 6. Confusion matrix for the RefineNet models.
Predicted | Winter wheat | Agricultural buildings | Woodland | Developed land | Roads | Water bodies | Farmland | Bare fields |
Winter wheat | 22.95% | 0.00% | 0.68% | 0.01% | 0.06% | 0.08% | 0.84% | 0.29% |
Agricultural buildings | 0.00% | 0.07% | 0.00% | 0.01% | 0.01% | 0.00% | 0.00% | 0.00% |
Woodland | 0.43% | 0.00% | 6.59% | 0.05% | 0.00% | 0.00% | 0.07% | 1.89% |
Developed land | 0.00% | 0.03% | 0.31% | 16.76% | 0.02% | 0.00% | 1.03% | 0.21% |
Roads | 0.01% | 0.01% | 0.00% | 0.12% | 0.61% | 0.00% | 0.02% | 0.06% |
Water bodies | 0.21% | 0.00% | 0.04% | 0.00% | 0.00% | 0.65% | 0.00% | 0.04% |
Farmland | 2.25% | 0.01% | 0.03% | 0.01% | 0.12% | 0.00% | 19.53% | 2.06% |
Bare fields | 0.35% | 0.00% | 0.03% | 0.39% | 0.01% | 0.00% | 1.67% | 19.38% |
In this study, we employed several criteria, including accuracy, precision, recall, and the Kappa coefficient, to evaluate the performance of the proposed model [52]. Table 7 shows data from the evaluation criteria of the three models, with the accuracy of the CEM model being 15.14% and 6.72% above those of SegNet and RefineNet, respectively.
Table 7. Comparison of the performance of the three models.
Index | CEM | SegNet | RefineNet |
Accuracy | 93.26% | 78.12% | 86.54% |
Precision | 91.75% | 74.38% | 80.87% |
Recall | 92.01% | 76.64% | 82.02% |
Kappa | 91.64% | 69.53% | 80.54% |
5. Discussion
5.1 Tuning of the parameter settings of the proposed CEM
When constructing the structure of the CEM model, we borrowed from the classical RefineNet model and the structure of the VGG model. The multipath structure of the RefineNet model can effectively integrate high- and low-level features and improve the feature extraction ability of the model. We implemented the multipath structure as the basic structure of the CEM model. Considering the characteristics of GF-2 remote sensing images, we set up a feature extraction unit using three downscaling and two feature extraction units that do not change scale. This design could ensure that the CEM model extracts enough features.
When designing the structure of the feature extraction unit, we borrowed the structure of the VGG model and adopted three convolution layer superposition strategies. The advantage of this design is it makes it possible to ensure that feature extraction has enough receptive domains and can reduce parameters.
Considering that there are a large number of mixed pixels in the GF-2 images, the use of spectral features is beneficial to improve classification accuracy. Two convolution kernels were set for each class to extract spectral features, and a total of 16 convolution kernels were set. Among them, considering that the infrared band in the GF-2 image is sensitive to vegetation, a fixed convolution kernel was set to map the infrared band's ability to respond to vegetation.
The above strategy effectively improved the generalization ability of the model.
5.2 The Advantages of CEM
We first analyzed the characteristics of the crop planting area in the GF-2 images and designed the structure of the CEM model based on those characteristics. By adopting two convolution kernels and utilizing the multi-path refinement structure, the network structure of the CEM model is well adapted to the characteristics of GF-2 images. This explains the higher precision obtained from the CEM model.
The field surveys revealed that in the crop planting area, a GF-2 pixel covers an area containing about 350 - 450 winter wheat or rice plants and about 180 maize plants. Since the content of each pixel area showed no significant difference and the texture is relatively smooth, the multi-layer convolution hardly generated more effective features. It even introduces noise, resulting in poor segmentation effects as presented in Tables 3 and 4, and Figure 6 and 7.
Considering the similarity in the structures of SegNet, RefineNet, and CEM, we introduced the difference between the maximum value of the category probability vector generated by the classifier and the next maximum value as an indicator to further illustrate the advantages of the CEM model structure. This indicator was termed category confidence. Figure 8 illustrates the category confidence distribution for the results of the experiment. The ratios of pixels at a lower confidence level of the comparison models are greater than those of the CEM model. This demonstrates that the feature value composition of the CEM model is superior to those of the SegNet and RefineNet models.
Figure 8. The category confidence distribution for the results of the experiment.
5.3 SegNet versus CEM
The SegNet model extracts high-level semantic features from rich image details through deep convolution. It demonstrates advantages when extracting objects that cover a large number of pixels. However, if the number of pixels covered by the object is small, or a pixel contains several target objects, this structure not only fails to extract detailed features, but may introduce additional noise due to the expansion of the field of view. This reduces the accuracy of the classification results from the model. When extracting crop spatial distributions from a GF-2 image, although the number of pixels occupied by a farmland is large, the difference between pixels remains small because the area covered by a plant is small, and this obscures the advantages of SegNet.
Contrary to the SegNet model, which extracts 64 high-level semantic features by deepening the hierarchy, the CEM model extracts 80 features by combining spectral features, high-level semantic features, and low-level semantic features. Because CEM fully considers the natural features of crops and the distribution characteristics of farmlands, it is advantageous for identifying pixels at the edges and corners of crop planting areas.
Figure 9 shows a comparison of the results of the two models. The images in Fig. 9 are dated February 20, 2017. As can be seen from Figure 9, only a few pixels are misclassified at the corners in case of the CEM results, while a significantly higher number of pixels are misclassified in the SegNet results, both at the corners and edges, and even within the planting area. In Regions 1 and 2, SegNet's results show more severe jaggedness, and our model has almost no jaggedness. In Regions 3, 4, 5, and 6, SegNet does not make a recognition at all, and our model’s recognition effect is very consistent with the actual regions, indicating that our model overcomes the defects of the SegNet model. Further, the use of only high-level semantic features is an important cause of edge roughness. The use of low-level semantic features and high-level features through fusion is an important way to improve the edge.
Figure 9. Segmentation error comparison: (a) original images, (b) ground truth, (c) errors of SegNet results, and (d) errors of CEM results.
5.4 RefineNet versus CEM
The RefineNet model also extracts high-level semantic features by deepening the hierarchy. It employs a multi-path refinement structure to fuse low- and high-level semantic features, compensating for missing information in down-sampling. The accuracy of its extraction results is therefore significantly better than that of the SegNet model. However, due to the noise introduced by the deep convolution structure, the edges of the crop planting area in the extraction result remain coarse.
Compared to camera images, remote sensing images reveal fewer details, but with a higher number of channels. Therefore, the structure of the CEM induces advantages including the division of the convolutional layer into five coding units and decoding them separately. The CEM model obtains sufficient information for classification processing while reducing the depth of the convolutional network, by using convolution kernels of 1×1 and 3×3 together. In addition to extracting different level semantic features of a pixel, the CEM model also extracts spectral features. These techniques ensure that the model produces high-precision results.
Figure 10 shows a comparison between the results of the two models. The images in Fig. 9 are dated March 16, 2018. As can be seen from Figure 10, only a few pixels are misclassified at the corners of the CEM results, while the RefineNet results show misclassified pixels at the corners and edges of planting areas. Although the number of misclassified pixels in the case of RefineNet is below that for SegNet, it is still significantly higher than that for CEM. In Regions 1, 2, and 4, the RefineNet model classification has many errors, resulting in rough edges, and in the CEM results show areas in more accurate detail. In Regions 3, 5, and 6, the RefineNet results contain more recognition errors and a considerable amount of winter wheat is not recognized. For the CEM results, the shape obtained is consistent with the actual shapes of areas. Considering the difference in the fusion of high-and low-level semantic features, the strategy adopted in this study is more reasonable than that of RefineNet.
Figure 10. Segmentation errors comparison: (a) original images, (b) ground truth, (c) errors of RefineNet results, and (d) errors of CEM results.
4. There is not clear the number of classes of the proposed problem, 8? 9? Page 4, lines 154-156: “Codes 1-8 were used to represent each land type in the label files for the training stage. In the label file produced by the CEM, code 1 denotes winter wheat, and code 9 denotes non-winter wheat.” Please explain better the number of classes of the problem.
Reply: According to your good suggestion, we have revised the relevant content to make the expression clear.
Firstly, we revised the relevant content in Section 2.3, The revised content is as follows:
We created a label file for each image to record its category number. Each pixel was assigned a code value in the label file. Codes 1-8 were used to represent each land type in the label files. The task of labeling each pixel was performed using the visual interpretation function of ENVI. Fig. 3 shows an image and its corresponding label.
Secondly, we added explanation in the revised Section 4.2. The revised content is as follows:
Figure 6 presents the extraction results of our model and those of two other models used for comparison with ours. Considering that the goal of this study was to extract crop planting areas, we use Figure 7 to show the results in which the other seven land use types are combined. We use code 9 to present the non-winter wheat.
5. Page 6 and following pages, explanations of the tuning of the parameter setting is not comprehensive. There an amount of arbitrary fixed values without explanation. A rationale about this should be included. Why five extraction units of semantic feature, 15 ordinary kernels, three feature extraction layers, 16 features…? Have those parameters been tuned to a problem of 9 classes? Explain the generalization capability of the proposed method to other problems.
Reply: According to your good suggestion, we revised the Section 3.1, added the Section 5.1, to make the tuning of parameter setting clear.
The revised Section 3.1 is as follows:
3.1 Structure of the proposed CEM
The CEM model consists of a feature extractor and classifier; the feature extractor comprises an encoder and decoder (Figure 4), The fused GF-2 image and the corresponding label file were used as input. The band order of the image is red, blue, green, and near-infrared.
Figure 4. Structure of the proposed CEM model. ReLU: rectified linear unit.
The encoder composed of a spectral feature extraction unit and five semantic feature extraction units. In Fig. 4, E0 represents the spectral feature extraction unit, and E1, E2, E3, E4, and E5 represent semantic feature extraction units, respectively. The word semantic means “similar category information”; in other words, the feature extraction unit extracts similar feature values from pixels belonging to the same category.
The spectral feature extraction unit uses 16 1×1-type convolution kernels, including a fixed convolution kernel and 15 ordinary convolution kernels, capable of extracting 16 features from the spectral information of a pixel. The value of the fixed convolution kernel is represented by the vector [0, 0, 0, x], and only the last component is adjusted during the training process. The purpose of the design is to fully exploit the sensitivity of the near-infrared band to vegetation. The image row and the column structure are unaltered by the spectral feature extraction unit. Therefore, we used this structure as the basic feature for the fusion of each feature level.
Each semantic feature extraction unit includes three feature extraction layers and one pooling layer. Each feature extraction layer has a convolutional layer, a batch normalization layer, and an activation layer for extracting semantic features. The convolutional layers comprise 3×3-type convolution kernels. Table 1 presents the number of convolution kernels for each convolutional layer. The activation layer uses the rectified linear unit function as the activation function.
Table 1. Number of convolution kernels for each convolutional layer.
Layer | Number of kernels |
1,2,3 | 64 |
4,5,6, | 128 |
7,8,9 | 256 |
10,11,12,13,14,15 | 512 |
The pooling layer was used to optimize the features, but because the ordinary 2×2 pooling kernel reduces the resolution of the image, we adopted a new pooling strategy. As there are a greater number of pixels for the same land use type in the early feature extraction stages, we used a 2×2 type pooling kernel in the 1st, 2nd, and 3rd feature extraction units. The use of the 2×2 type pooling kernel potentially accelerates feature aggregation. In the later stages of feature extraction, the resolution differs significantly from that of the original image, so we adjusted the step size to 1 in the 4th and 5th units. In the 4th and 5th feature extraction units, the size of the pooling kernel is still 2×2, but when the size of the feature block participating in the pooling operation is smaller than the size of the pooling kernel, the size of the pooling kernel is adjusted to match the feature block; this ensures that valid pooling values are obtained. Therefore, the 4th and 5th feature extraction units do not change the size of the feature map.
The decoder is also composed of five units, as shown by D1, D2, D3, D4, and D5 in Fig. 5. The decoder up-samples the feature map to gradually restore the feature image to the size of the original image. The decoder generates feature vectors with the same structure for each pixel.
The D5 and D4 units are composed of convolution layers. The convolution layer, which works in the ‘same’ mode in Tensonflow, is used to adjust feature values without changing the size of the feature map. The D5 unit directly uses the output of the E5 unit as input. The input of the D4 unit is the output of the D5 unit and the output of the E4 unit. The inputs are first fused using Equation 1 and then adjusted using the convolution layer.
(1)
In Equation 1, f denotes the fused feature map, d5 denotes the output of the D5 unit, and e4 denotes the output of the E4 unit. A and b are fusion coefficients, and the values of a and b are determined after training.
The convolution layer of D4 adjust the depth of the fused feature map to 256. The D3 unit is composed of an up-sampling layer and a convolutional layer. The up-sampling layer is used to up sample the output of the D4 unit to match the size of output of the E3 unit; Equation 1 is then used for fusion, but use fusion coefficients is different from those used in D4. The fused features are also adjusted through the convolutional layer. The structures of the D2 and D1 units are the same as that of the D3 unit, and the same working mode is adopted. Table 2 presents the depth of feature map generated by each decoder unit.
Table 2. Depth of feature map generated by each decoder unit.
Decoder unit | Depth of feature map |
D5 | 512 |
D4, | 512 |
D3 | 256 |
D2 | 128 |
D1 | 64 |
Finally, the semantic features generated by the D1 unit are concatenated with the spectral features outputted by E0 to form a feature map.
The CEM model uses the SoftMax model as a classifier. The SoftMax model exploits the fused feature map generated by the feature extractor to produce the category probability vector for each pixel, and then uses the category corresponding to the maximum value of the probability vector as the category of the pixel.
The added Section 5.1 is as follows:
5.1 Tuning of the parameter settings of the proposed CEM
When constructing the structure of the CEM model, we borrowed from the classical RefineNet model and the structure of the VGG model. The multipath structure of the RefineNet model can effectively integrate high- and low-level features and improve the feature extraction ability of the model. We implemented the multipath structure as the basic structure of the CEM model. Considering the characteristics of GF-2 remote sensing images, we set up a feature extraction unit using three downscaling and two feature extraction units that do not change scale. This design could ensure that the CEM model extracts enough features.
When designing the structure of the feature extraction unit, we borrowed the structure of the VGG model and adopted three convolution layer superposition strategies. The advantage of this design is it makes it possible to ensure that feature extraction has enough receptive domains and can reduce parameters.
Considering that there are a large number of mixed pixels in the GF-2 images, the use of spectral features is beneficial to improve classification accuracy. Two convolution kernels were set for each class to extract spectral features, and a total of 16 convolution kernels were set. Among them, considering that the infrared band in the GF-2 image is sensitive to vegetation, a fixed convolution kernel was set to map the infrared band's ability to respond to vegetation.
The above strategy effectively improved the generalization ability of the model.
6. Several technical aspects should be addressed: (i) the value of the cost function through iterations should be represented and explained. (ii) Table 2, values of confusion matrices must add to 100%. (iii) The “semantic” features should be represented and discussed. (iv) The discussion of Section 5 should include more supporting numerical data, for instance, page 10, lines 297-299, “more high-level semantic features”, “”more features”,…
Reply: According to your good suggestion, we revised the relevant content. The point-by-point response to what you have highlighted is as follows.
(i) the value of the cost function through iterations should be represented and explained.
We added the content in Section 4.2. The new content is as follows:
Figure 5 presents the CEM loss rate curve for training. As can be seen from the figure, as the number of iterations increases, the loss value decreases and it eventually tends to stabilize. The loss value decreases rapidly during the first 5000 iterations. From iteration 5000 to 15000, the loss value fluctuates. After 15000 iterations, the loss value tends to become stable. The initial learning rate set in the training phase is 0.0001, and the learning rate is reduced to half the existing value every 5000 iterations. It can be seen that as the learning rate decreases, the training loss value also slightly decreases within a certain range. This proves that the learning rate gradient has benefit effect on the training network.
Figure 5. Loss rate curve.
(ii) Table 2, values of confusion matrices must add to 100%.
We have check the values of confusion matrices, and correct the mistakes. The revised Table 2, now named Table 4, Table 5, Table 6, is as follows:
Table 4. Confusion matrix for the CEM models.
Predicted | Winter wheat | Agricultural buildings | Woodland | Developed land | Roads | Water bodies | Farmland | Bare fields |
Winter wheat | 23.15% | 0.00% | 0.16% | 0.03% | 0.01% | 0.08% | 0.97% | 0.51% |
Agricultural buildings | 0.00% | 0.08% | 0.00% | 0.01% | 0.00% | 0.00% | 0.00% | 0.00% |
Woodland | 0.05% | 0.00% | 8.87% | 0.03% | 0.01% | 0.01% | 0.03% | 0.03% |
Developed land | 0.02% | 0.01% | 0.00% | 17.37% | 0.02% | 0.00% | 0.53% | 0.41% |
Roads | 0.02% | 0.00% | 0.00% | 0.04% | 0.72% | 0.00% | 0.02% | 0.03% |
Water bodies | 0.09% | 0.00% | 0.01% | 0.00% | 0.00% | 0.83% | 0.01% | 0.00% |
Farmland | 0.38% | 0.00% | 0.15% | 0.38% | 0.02% | 0.00% | 22.18% | 0.90% |
Bare fields | 0.33% | 0.00% | 0.36% | 0.19% | 0.02% | 0.00% | 0.87% | 20.06% |
Table 5. Confusion matrix for the Segnet models.
Predicted | Winter wheat | Agricultural buildings | Woodland | Developed land | Roads | Water bodies | Farmland | Bare fields |
Winter wheat | 20.82% | 0.00% | 1.13% | 1.80% | 0.06% | 0.06% | 0.77% | 0.27% |
Agricultural buildings | 0.01% | 0.05% | 0.00% | 0.01% | 0.01% | 0.00% | 0.00% | 0.01% |
Woodland | 0.85% | 0.00% | 6.25% | 0.00% | 0.00% | 0.01% | 0.91% | 1.01% |
Developed land | 0.73% | 0.04% | 0.00% | 14.47% | 0.11% | 0.00% | 1.15% | 1.86% |
Roads | 0.03% | 0.00% | 0.00% | 0.01% | 0.69% | 0.00% | 0.09% | 0.01% |
Water bodies | 0.09% | 0.00% | 0.05% | 0.03% | 0.00% | 0.67% | 0.08% | 0.02% |
Farmland | 0.56% | 0.00% | 0.01% | 0.01% | 0.12% | 0.00% | 18.34% | 4.97% |
Bare fields | 1.19% | 0.00% | 0.00% | 0.02% | 0.02% | 0.00% | 3.77% | 16.83% |
Table 6. Confusion matrix for the RefineNet models.
Predicted | Winter wheat | Agricultural buildings | Woodland | Developed land | Roads | Water bodies | Farmland | Bare fields |
Winter wheat | 22.95% | 0.00% | 0.68% | 0.01% | 0.06% | 0.08% | 0.84% | 0.29% |
Agricultural buildings | 0.00% | 0.07% | 0.00% | 0.01% | 0.01% | 0.00% | 0.00% | 0.00% |
Woodland | 0.43% | 0.00% | 6.59% | 0.05% | 0.00% | 0.00% | 0.07% | 1.89% |
Developed land | 0.00% | 0.03% | 0.31% | 16.76% | 0.02% | 0.00% | 1.03% | 0.21% |
Roads | 0.01% | 0.01% | 0.00% | 0.12% | 0.61% | 0.00% | 0.02% | 0.06% |
Water bodies | 0.21% | 0.00% | 0.04% | 0.00% | 0.00% | 0.65% | 0.00% | 0.04% |
Farmland | 2.25% | 0.01% | 0.03% | 0.01% | 0.12% | 0.00% | 19.53% | 2.06% |
Bare fields | 0.35% | 0.00% | 0.03% | 0.39% | 0.01% | 0.00% | 1.67% | 19.38% |
(iii) The “semantic” features should be represented and discussed.
We have added new content to represent the term “semantic” in Section 3.1. The new content is as follows:
The encoder composed of a spectral feature extraction unit and five semantic feature extraction units. In Fig. 4, E0 represents the spectral feature extraction unit, and E1, E2, E3, E4, and E5 represent semantic feature extraction units, respectively. The word semantic means “similar category information”; in other words, the feature extraction unit extracts similar feature values from pixels belonging to the same category.
(iv) The discussion of Section 5 should include more supporting numerical data, for instance, page 10, lines 297-299, “more high-level semantic features”, “”more features”,…
We revised the relevant content, and the revised content is as follows:
Contrary to the SegNet model, which extracts 64 high-level semantic features by deepening the hierarchy, the CEM model extracts 80 features by combining spectral features, high-level semantic features, and low-level semantic features.

Reviewer 4 Report
This paper seeks to extract fine crop spatial distribution information based on the characteristics of the crop planting area in Gaofen 2 (GF-2) images. They established a convolutional neural network consisting of a feature extractor and a classifier for crops.
Overall, I do see that this approach can be better than other models but the paper is hard to follow. For example, the abstract states “The classifier utilizes fused features generated by the feature extractor to perform pixel-by-pixel classifications.” I’m not sure what these fused features are, I’m assuming the multiple images and corresponding bands. Additionally, I’m not seeing a huge difference in model architecture in Figure 5 than other popular approaches (SegNet, etc.). Maybe I’m not following becausae it’s hard to follow. It would be nice to add more text to Figure 5 or a Table for the comparison models, highlighting the differences.
Is Table 2 comparing the test images or the ground-truth data (section 2.3)? It would be good to add a Table on the datasets used for train, test, and validation.
Author Response
Response to reviewer's comments from Applied Sciences (applsci-524108)
Dear Reviewer:
We would like to thank you for the comments and suggestion. We have substantially revised the manuscript and detailed responses are provided below. All revised contents are in blue.
1. Overall, I do see that this approach can be better than other models but the paper is hard to follow. For example, the abstract states “The classifier utilizes fused features generated by the feature extractor to perform pixel-by-pixel classifications.” I’m not sure what these fused features are, I’m assuming the multiple images and corresponding bands. Additionally, I’m not seeing a huge difference in model architecture in Figure 5 than other popular approaches (SegNet, etc.). Maybe I’m not following becausae it’s hard to follow. It would be nice to add more text to Figure 5 or a Table for the comparison models, highlighting the differences.
Reply: According to your good suggestion, we revised the relevant content. Firstly, we revised the abstract. Secondly, we revised the Section 3.1. to make the CEM model clear. The revised content is as follows.
The revised abstract is as follows:
Using satellite remote sensing has become a mainstream approach for extracting crop spatial distribution. Making edges finer is a challenge while extracting crop spatial distribution information from high-resolution remote sensing images using a convolutional neural network (CNN). Based on the characteristics of the crop area in the Gaofen 2 (GF-2) images, this paper proposes an improved CNN to extract fine crop areas. The CNN comprises a feature extractor and a classifier. The feature extractor employs a spectral feature extraction unit to generate spectral features, and five coding-decoding-pair units to generate five level features. A linear model is used to fuse features of different levels, and the fusion results are up-sampled to obtain a feature map consistent with the structure of the input image. This feature map is used by the classifier to perform pixel-by-pixel classification. In this study, the SegNet and RefineNet models and 21 GF-2 images of Feicheng County, Shandong Province, China, were chosen for comparison experiment. Our approach had an accuracy of 93.26%, which is higher than those of the existing SegNet (78.12%) and RefineNet (86.54%) models. This demonstrates the superiority of the proposed method in extracting crop spatial distribution information from GF-2 remote sensing images.
The revised Section 3.1 is as follows:
3.1 Structure of the proposed CEM
The CEM model consists of a feature extractor and classifier; the feature extractor comprises an encoder and decoder (Figure 4), The fused GF-2 image and the corresponding label file were used as input. The band order of the image is red, blue, green, and near-infrared.
Figure 4. Structure of the proposed CEM model. ReLU: rectified linear unit.
The encoder composed of a spectral feature extraction unit and five semantic feature extraction units. In Fig. 4, E0 represents the spectral feature extraction unit, and E1, E2, E3, E4, and E5 represent semantic feature extraction units, respectively. The word semantic means “similar category information”; in other words, the feature extraction unit extracts similar feature values from pixels belonging to the same category.
The spectral feature extraction unit uses 16 1×1-type convolution kernels, including a fixed convolution kernel and 15 ordinary convolution kernels, capable of extracting 16 features from the spectral information of a pixel. The value of the fixed convolution kernel is represented by the vector [0, 0, 0, x], and only the last component is adjusted during the training process. The purpose of the design is to fully exploit the sensitivity of the near-infrared band to vegetation. The image row and the column structure are unaltered by the spectral feature extraction unit. Therefore, we used this structure as the basic feature for the fusion of each feature level.
Each semantic feature extraction unit includes three feature extraction layers and one pooling layer. Each feature extraction layer has a convolutional layer, a batch normalization layer, and an activation layer for extracting semantic features. The convolutional layers comprise 3×3-type convolution kernels. Table 1 presents the number of convolution kernels for each convolutional layer. The activation layer uses the rectified linear unit function as the activation function.
Table 1. Number of convolution kernels for each convolutional layer.
Layer | Number of kernels |
1,2,3 | 64 |
4,5,6, | 128 |
7,8,9 | 256 |
10,11,12,13,14,15 | 512 |
The pooling layer was used to optimize the features, but because the ordinary 2×2 pooling kernel reduces the resolution of the image, we adopted a new pooling strategy. As there are a greater number of pixels for the same land use type in the early feature extraction stages, we used a 2×2 type pooling kernel in the 1st, 2nd, and 3rd feature extraction units. The use of the 2×2 type pooling kernel potentially accelerates feature aggregation. In the later stages of feature extraction, the resolution differs significantly from that of the original image, so we adjusted the step size to 1 in the 4th and 5th units. In the 4th and 5th feature extraction units, the size of the pooling kernel is still 2×2, but when the size of the feature block participating in the pooling operation is smaller than the size of the pooling kernel, the size of the pooling kernel is adjusted to match the feature block; this ensures that valid pooling values are obtained. Therefore, the 4th and 5th feature extraction units do not change the size of the feature map.
The decoder is also composed of five units, as shown by D1, D2, D3, D4, and D5 in Fig. 5. The decoder up-samples the feature map to gradually restore the feature image to the size of the original image. The decoder generates feature vectors with the same structure for each pixel.
The D5 and D4 units are composed of convolution layers. The convolution layer, which works in the ‘same’ mode in Tensonflow, is used to adjust feature values without changing the size of the feature map. The D5 unit directly uses the output of the E5 unit as input. The input of the D4 unit is the output of the D5 unit and the output of the E4 unit. The inputs are first fused using Equation 1 and then adjusted using the convolution layer.
(1)
In Equation 1, f denotes the fused feature map, d5 denotes the output of the D5 unit, and e4 denotes the output of the E4 unit. A and b are fusion coefficients, and the values of a and b are determined after training.
The convolution layer of D4 adjust the depth of the fused feature map to 256. The D3 unit is composed of an up-sampling layer and a convolutional layer. The up-sampling layer is used to up sample the output of the D4 unit to match the size of output of the E3 unit; Equation 1 is then used for fusion, but use fusion coefficients is different from those used in D4. The fused features are also adjusted through the convolutional layer. The structures of the D2 and D1 units are the same as that of the D3 unit, and the same working mode is adopted. Table 2 presents the depth of feature map generated by each decoder unit.
Table 2. Depth of feature map generated by each decoder unit.
Decoder unit | Depth of feature map |
D5 | 512 |
D4, | 512 |
D3 | 256 |
D2 | 128 |
D1 | 64 |
Finally, the semantic features generated by the D1 unit are concatenated with the spectral features outputted by E0 to form a feature map.
The CEM model uses the SoftMax model as a classifier. The SoftMax model exploits the fused feature map generated by the feature extractor to produce the category probability vector for each pixel, and then uses the category corresponding to the maximum value of the probability vector as the category of the pixel.
2. Is Table 2 comparing the test images or the ground-truth data (section 2.3)? It would be good to add a Table on the datasets used for train, test, and validation.
Reply: According to your good suggestion, we revised the Section 4.1, added new content of dataset. The revised content is as follows:
4.1 Experimental Setup
The Python 3.6 and the TensorFlow framework were employed on a Linux Ubuntu 16.04 operating system to implement the proposed CEM model. A workstation equipped with a 12 GB NVIDIA Graphics card was used to perform the comparison experiments.
SegNet [48] and RefineNet [51] are state-of-the-art classic pixel-by-pixel semantic image segmentation models for camera images. As the working principles of SegNet and RefineNet are similar to that of the CEM model, we employed these as comparison models to better reflect the superiority of feature extraction and classification of our model.
Since the main crop in Feicheng County during winter is winter wheat, we selected the winter wheat area as the target for experiments.
Total 375 sub-images were selected from the 537 images described in Section 2.3 to compose the training data set, 106 as validation data set, and the remaining 56 as test data set. The training data set, validation set, and test set include all land use types, respectively. Every image in the training data set was processed with color adjustment, horizontal flip, and vertical flip amongst others, to increase the number and diversity of the samples. The color adjustment factors included brightness, saturation, hue, and contrast. After this preprocessing, the final training dataset comprised 4,125 images. Table 3 shows the number of samples used in the experiment.
Table 3. Number of samples used in the experiment.
Category | Number of samples in training data set(million) | Number of samples in validation data set(million) | Number of samples in test data set (million) |
Winter wheat | 710 | 18 | 9 |
Agricultural buildings | 2 | 0.1 | 0.07 |
Woodland | 256 | 6 | 3 |
Developed land | 542 | 13 | 7 |
Roads | 23 | 0.6 | 0.3 |
Water bodies | 25 | 0.7 | 0.3 |
Farmland | 687 | 17 | 9 |
Bare fields | 602 | 15 | 8 |

Round 2
Reviewer 1 Report
Brilliant!
Congratulations!
Author Response
Dear Reviewer:
We would like to thank you for the review for our manuscript.

Reviewer 3 Report
The quality of the contents of the paper have improved significantly. The revised version of the paper shows a great effort has been made to respond all my previous comments satisfactorily. Thus, I consider the paper should be ready for publication.
Author Response

(The authors gave the same response as above.)

Reviewer 4 Report
Overall, it is an improved manuscript but still hard to follow the main differences between this approach and the other models (SegNet, etc.). The results look good but not sure how I would implement this model, as I worked with SegNet before in TensorFlow. I would suggest adding a section highlighting the main differences. I would also suggest adding more text to the Figures, for example Figure 4 has abbreviations of E, D, and F which need to explained in the caption.
Author Response
Dear Reviewer:
We would like to thank you for the comments and suggestion. We have substantially revised the manuscript and detailed responses are provided below. All revised contents are in blue.
1. Overall, it is an improved manuscript but still hard to follow the main differences between this approach and the other models (SegNet, etc.). The results look good but not sure how I would implement this model, as I worked with SegNet before in TensorFlow. I would suggest adding a section highlighting the main differences. I would also suggest adding more text to the Figures, for example Figure 4 has abbreviations of E, D, and F which need to explained in the caption.
Reply: According to your good suggestion, we revised the relevant content. Firstly, we added new content highlighting the main differences between SegNet and CEM in Section 5.3. Secondly, we revised the Figure 4.
The revised Section 5.3.is as follows:
5.3 SegNet versus CEM
The SegNet model extracts high-level semantic features from rich image details through deep convolution. It demonstrates advantages when extracting objects that cover a large number of pixels. However, if the number of pixels covered by the object is small, or a pixel contains several target objects, this structure not only fails to extract detailed features, but may introduce additional noise due to the expansion of the field of view. This reduces the accuracy of the classification results from the model. When extracting crop spatial distributions from a GF-2 image, although the number of pixels occupied by a farmland is large, the difference between pixels remains small because the area covered by a plant is small, and this obscures the advantages of SegNet.
Contrary to the SegNet model, which extracts 64 high-level semantic features by deepening the hierarchy, the CEM model extracts 80 features by combining spectral features, high-level semantic features, and low-level semantic features. Because CEM fully considers the natural features of crops and the distribution characteristics of farmlands, it is advantageous for identifying pixels at the edges and corners of crop planting areas.
In summary, there are the following differences between CEM and SegNet model.
(1) The SegNet model transforms the highest-level semantic features into pixel feature vectors by step-by-step sampling. Therefore, pixel feature vectors only contain abstract semantic information. The CEM model adopts feature fusion to fuse low-level semantic feature information and high-level semantic information, so the information contained in feature vectors of pixels is more abundant than that of SegNet.
(2) The feature vector generated by the SegNet model has only semantic information. the feature vector generated by the CEM model not only has semantic information, but also statistical information of the spectral values of the pixels themselves.
(3) Although the pooling method adopted by SegNet has the effect of aggregating feature values, but each pooling reduces the size of feature map to 1/4 of the original size, which is not conducive to the generation of pixel feature vectors. The pooling method adopted by CEM not only achieves the purpose of aggregation of feature values, but also reduces the degree of feature map size reduction, which can help to obtain feature vectors with good discrimination.
The revised Figure 4 is as follows:
3.1 Structure of the proposed CEM
The CEM model consists of a feature extractor and classifier; the feature extractor comprises an encoder and decoder (Figure 4), The fused GF-2 image and the corresponding label file were used as input. The band order of the image is red, blue, green, and near-infrared.
Figure 4. Structure of the proposed CEM model. ReLU: rectified linear unit; D: Decoder; E: Encoder.
